# Illuminating the mechanism and allosteric behavior of NanoLuc luciferase

Michal Nemergut[1,2,7,8], Daniel Pluskal[1,8], Jana Horackova[1,8], Tereza Sustrova[1,8], Jan Tulis[1], Tomas Barta[3], Racha Baatallah[4], Glwadys Gagnot[4,5], Veronika Novakova[1,2], Marika Majerova[1,2], Karolina Sedlackova[1,2], Sérgio M. Marques[1,2], Martin Toul[1,2], Jiri Damborsky[1,2], Zbynek Prokop[1,2], David Bednar[1,2] ✉, Yves L. Janin[6] ✉ & Martin Marek[1,2] ✉

NanoLuc, a superior β-barrel fold luciferase, was engineered 10 years ago but the nature of its catalysis remains puzzling. Here experimental and computational techniques are combined, revealing that imidazopyrazinone luciferins bind to an intra-barrel catalytic site but also to an allosteric site shaped on the enzyme surface. Structurally, binding to the allosteric site prevents simultaneous binding to the catalytic site, and vice versa, through concerted conformational changes. We demonstrate that restructuration of the allosteric site can boost the luminescent reaction in the remote active site. Mechanistically, an intra-barrel arginine coordinates the imidazopyrazinone component of luciferin, which reacts with $O_2$ via a radical charge-transfer mechanism, and then it also protonates the resulting excited amide product to form a light-emitting neutral species. Concomitantly, an aspartate, supported by two tyrosines, fine-tunes the blue color emitter to secure a high emission intensity. This information is critical to engineering the next-generation of ultrasensitive bioluminescent reporters.

Bioluminescence is a fascinating phenomenon involving the emission of light by a living organism[1]. Hence, there is a huge interest in harnessing bioluminescent systems not only to design ultrasensitive optical bioassays but also to enable sustainable and environmentally friendly lighting technologies[2–5]. Bioluminescent systems all generate "cold light" via the oxidation of a substrate (a luciferin), which is catalyzed by a group of enzymes called luciferases[1].

In 1978, Shimomura et al. focused on the bioluminescence of *Oplophorus gracilirostris*, a deep shrimp that ejects a cloud of brightly luminescent secretion from the base of its antennae as a defense mechanism against predation[6]. The identified *O. gracilirostris* luciferase, henceforth referred to as OLuc, has a quaternary structure composed of two ~35 kDa and two ~19 kDa subunits[7]. As with many marine luciferases, it oxidizes coelenterazine (CTZ), an imidazopyrazinone containing luciferin, into a coelenteramide (CEI), in a cofactor-independent decarboxylating reaction to generate blue light ($\lambda_{max}$ ~ 460 nm)[6]. Cloning experiments showed that bioluminescent activity entirely relies on the smaller 19 kDa subunit[7]. Unfortunately, when this subunit is recombinantly produced alone, it does not retain many of the desirable properties evident in the native enzyme, as it is

[1]Loschmidt Laboratories, Department of Experimental Biology and RECETOX, Faculty of Science, Masaryk University, Kamenice 5, Bld. C13, 625 00, Brno, Czech Republic. [2]International Clinical Research Center, St. Anne's University Hospital Brno, Pekarska 53, 656 91, Brno, Czech Republic. [3]Department of Histology and Embryology, Faculty of Medicine, Masaryk University, Kamenice 753/5, 625 00, Brno, Czech Republic. [4]Unité de Chimie et Biocatalyse, Institut Pasteur, UMR 3523, CNRS, 28 rue du Dr. Roux, 75724 Paris Cedex 15, Paris, France. [5]Université de Paris, 12 rue de l'école de Médecine, 75006 Paris, France. [6]Structure et Instabilité des Génomes (StrInG), Muséum National d'Histoire Naturelle, INSERM, CNRS, Alliance Sorbonne Université, 75005 Paris, France. [7]Present address: Center for Interdisciplinary Biosciences, Technology and Innovation Park, P. J. Safarik University in Kosice, Trieda SNP 1, 04011 Kosice, Slovakia. [8]These authors contributed equally: Michal Nemergut, Daniel Pluskal, Jana Horackova, Tereza Sustrova. ✉e-mail: davidbednar1208@gmail.com; yves.janin@cnrs.fr; martin.marek@recetox.muni.cz

unstable and poorly soluble[7]. Therefore, structural optimization of the catalytic subunit involving extensive protein engineering was performed to create an innovative luciferase named NanoLuc (or NLuc), hand in hand with the design of a novel imidazopyrazinone substrate called furimazine (FMZ)[8].

While this engineered NanoLuc luciferase still catalyzes native CTZ-to-CEI reaction, it displays superior specific activity for the FMZ to furimamide (FMA) oxidation, leading to up to a 150-fold stronger light signal than those observed for firefly luciferase (FLuc) and *Renilla* luciferase (RLuc)[8]. NanoLuc is a multipurpose technology that is triggering a revolution in bioimaging, protein–protein or protein–ligand interaction studies, gene regulation and cell signaling, protein stability monitoring as well as the development of bioluminescence resonance energy transfer (BRET)-based sensors[4,9–15]. However, despite its incredible technological and commercial success[8,10,11], the nature of its luciferin-binding site and mechanism by which it generates blue photons remain unknown. In addition, the major drawbacks of the Nano-Luc system are that FMZ-luciferin is poorly soluble, possesses cytotoxic properties[16] and is substantially more expensive than widely accessible CTZ-luciferin.

In 2016, Tomabechi et al. determined the crystal structure of apo-NanoLuc[17]. The structure consists of eleven antiparallel β-strands (S1-11) forming a β-barrel that is capped by 4 α-helices (H1-4), displaying structural similarity with distantly related fatty acid-binding proteins (FABPs)[17]. The engineered NanoLuc, unlike the native OLuc, is reported as a monomeric enzyme, and it is anticipated that a luciferin binds to a central cavity of the β-barrel structure, where catalysis should occur[17–19]. In the meantime, three additional crystal structures were determined and deposited in the PDB database[20] (PDB ID codes: 7MJB, 5IBO and 7VSX[19]), two complexed with decanoic acid and one with 2-(*N*-morpholino)ethanesulfonic acid (MES). Apparently, a key barrier for further deciphering of the puzzling mechanism of NanoLuc catalysis is the unavailability of structural data depicting luciferin-bound enzyme complexes.

Here we determine co-crystal structures of NanoLuc luciferase complexed with oxidized imidazopyrazinone oxyluciferins as well as a non-oxidizable substrate analog azacoelenterazine[21]. We demonstrate that the luciferins can bind not only to an intra-barrel catalytic site but also to a secondary, allosteric site localized on the molecular surface of the enzyme. Binding to the allosteric site prevents simultaneous binding to the catalytic site through the so-called homotropic negative allostery mechanism. Moreover, we reveal molecular details of Nano-Luc catalytic machinery and delineate its reaction mechanism. All these mechanistic insights should be critical to engineering the next generation of luciferin/luciferase reporting systems and renewable light-producing technologies.

## Results
### Identification of a luciferin-binding site on the enzyme surface
We found new crystallization conditions leading to diffraction-quality NanoLuc crystals, which were then extensively soaked in the luciferin-supplemented mother liquor. Diffraction experiments yielded high-resolution data, and the structures were solved by molecular replacement (Supplementary Table 1). Notably, the NanoLuc is packed in the crystals as a back-to-back dimer of two homodimers (crystallographic homotetramer) with a central pore, where all four carboxy-terminal ends are involved in the self-association. We identified a chloride ion that occupies the central pore, and it thus contributes to the homotetrameric association (Supplementary Fig. 1). Strikingly, apart from intramolecular cavities, the crystal packing also revealed several spacious pockets shaped on the molecular surface of NanoLuc, which might potentially serve as a luciferin-binding site (Supplementary Fig. 1).

In fact, inspection of the electron density maps unambiguously revealed FMA, an amide product of FMZ oxidation (Supplementary

Fig. 2), bound to a voluminous pocket found on the molecular surface of the enzyme, and localized at the crystallographic homodimer interface (Fig. 1a). Strikingly, while the previously determined decanoic acid-bound NanoLuc crystal structures are composed of four symmetry-related monomers leading to a perfect symmetric homo-dimer (PDB ID codes 7MJB and 5IBO), our luciferin-soaked structures revealed a symmetry breaking in the dimer interface. The origin of this interface asymmetry lies in the rearrangement of a structural element encompassing a helix H4, a loop L7, and a strand S4 (Fig. 1b, c). The B-factor analysis demonstrated that both the helix H4 and adjacent loop L7, unlike the β-barrel core, are highly mobile elements (Fig. 1d). PISA calculations[22] showed that the crystallographic dimer interface area is ~830 Å², which represents ~9.2% of the total solvent-accessible surface area of the monomer (~8995 Å²). There are 27 (chain A) and 28 (chain B) interfacing residues involved in the dimer interface, which is a significant portion of 169-residue protein (Fig. 1e, f).

### Nature of the surface-localized luciferin-binding site
The luciferin-binding surface pocket is defined by the groove on the β-barrel surface of one monomer (chain A), shaped by long strands S3 and S5, building the pocket bottom (Fig. 1g, h). The sides of the pocket are formed by strands S1 and S4 of the same monomer. The lid of the pocket is secured by an amino-terminal part, predominantly residues E4, V7, G8 and D9, of a second monomer (chain B). The back side of the pocket is closed by the carboxy-terminal tails (residues 166–169) of both chains (Fig. 1h).

FMA-oxyluciferin adopts a crab-like conformation, where the $R^1$ 2-(furan-2-yl) and $R^3$ 8-benzyl substituents constitute the "claws", while the $R^2$ 6-phenyl substituent is the tail (Fig. 1g). The latter part, the 6-phenyl moiety, is deeply buried in the pocket, where it is anchored through multiple non-polar and hydrophobic contacts with D9, I41 and I167 (Fig. 1h). The 8-benzyl substituent makes hydrophobic contacts with V7 (chain B) and V83 (chain A). The 2-(furan-2-yl) is positioned in proximity to a tyrosine-tyrosine dyad, allowing T-shaped π-stacking with Y81 (4.5 Å), and hydrophobic contacts with Y94 (4.3 Å). Moreover, the 2-(furan-2-yl) makes a π-cation interaction with the side chain of K89.

The acetamidopyrazine core is shielded by the side chain of R43, which at the same time makes both a bidentate hydrogen bonding with the carboxylate of D9 and a hydrogen bonding with the main-chain carbonyl of G8, provided by the second monomer (chain B). Crucially, the FMA carbonyl oxygen makes a hydrogen bond with the carboxylate of D55 (2.4 Å), while the amide nitrogen is hydrogen bonded with the side chain of K89 (2.4 Å) (Fig. 1h). As shown in Fig. 1i, several pocket-shaping residues (e.g., E4, R43 and R166), were introduced during the design of NanoLuc[8].

### One pocket, two radically different luciferin-binding modes
While the FMA adopts crab-like conformation, employing the tail part to be inserted in the luciferin-binding pocket (Fig. 1g), a radically different binding mode is observed for the CEI-oxyluciferin (Fig. 2a–c). Compared to the FMA-oxyluciferin, the CEI is horizontally rotated by ~120°, allowing its $R^1$ 2-(p-hydroxyphenyl)acetamide moiety to be deeply buried in the pocket. Though the FMA-luciferin makes inter-actions only within the asymmetric dimer (chains A and B), the hydroxyl group of CEI $R^1$ 2-(p-hydroxyphenyl) makes a hydrogen bond (3.4 Å) with the carboxylate of E165 in neighboring chain C (Fig. 2d). In addition, the CEI carbonyl group forms a hydrogen bond with the carboxylate of D9 (3.1 Å), and the nitrogen atoms of the central pyrazine ring are hydrogen bonded with the side chains of K89 (2.9 Å) and R166 (2.5 Å). The $R^2$ 6-(p-hydroxyphenyl) moiety makes contacts with Y81 and D5 (Fig. 2d). The superposition of FMA and CEI-binding modes demonstrates that both luciferins bind to the same pocket, but in a dissimilar fashion (Fig. 2e), highlighting luciferin-specific molecular recognition determinants. Moreover, the luciferin-binding site

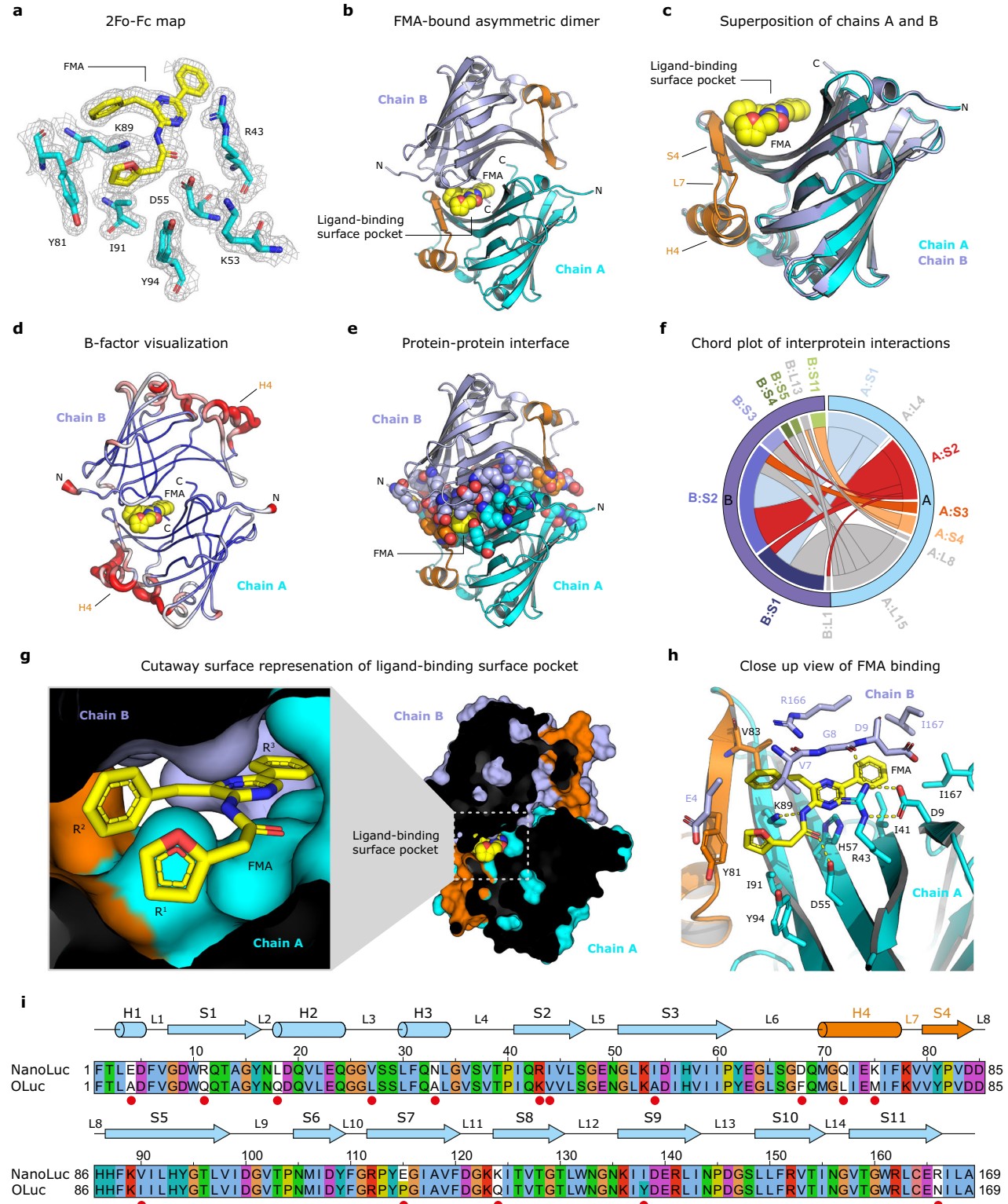

**Fig. 1 | The structure of FMA bound in the ligand-binding surface pocket of NanoLuc luciferase. a** 2Fo-Fc electron density (contour level 1.2 σ) at the FMA-oxyluciferin binding site. **b** Cartoon representation of the overall structure of NanoLuc asymmetric dimer (chain A in cyan and chain B in blue) with bound FMA luciferin (yellow). **c** Superposition of chain A (cyan) and chain B (blue). The structural element responsible for symmetry breaking, encompassing helix H4, loop L7 and strand S4, is colored orange. **d** B-factor putty representation of NanoLuc homodimer. **e** Dimer interface visualization. All residues involved in the dimer interface are shown as space-filling spheres. FMA is shown as yellow spheres. **f** Chord plot showing the interactions between chains A and B in the NanoLuc

dimer at the secondary structure level, calculated and visualized by Protein Contact Atlas[83]. **g** Cutaway surface representation of FMA-bound NanoLuc dimer. **h** Close-up view of FMA-binding pocket with residues creating the active site in stick representation. Key hydrogen bonds are shown as dashed yellow lines. **i** Sequence alignment between NanoLuc and the catalytic unit of *O. gracilirostris* luciferase (OLuc). Secondary structure elements found in NanoLuc are shown above the alignment. Amino acid residues mutated during the NanoLuc engineering[8] are labeled with the red dot. The numbering indicated above the alignment corresponds to the NanoLuc structure (PDB ID: 5B0U)[17].

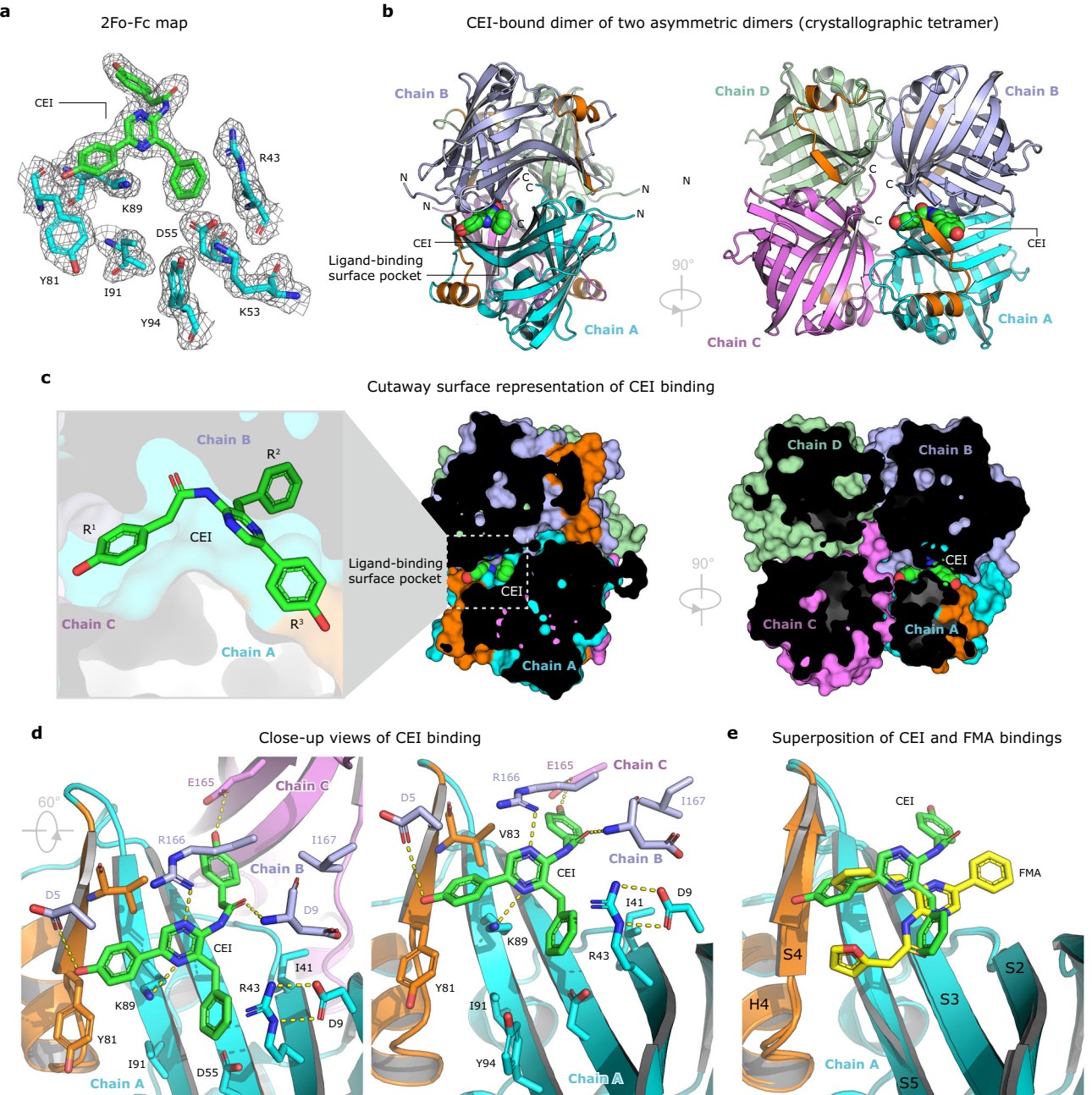

**Fig. 2 | The structure of CEI-bound crystallographic homotetramer of NanoLuc luciferase. a** 2Fo-Fc electron density (contour level 1.2 σ) at the CEI-oxyluciferin binding site. **b** The overall structure of the NanoLuc homotetramer, composed of two tail-to-tail asymmetric homodimers (chain A in cyan, chain B in blue, chain C in violet and chain D in green). The CEI luciferin is shown as space-filling spheres (green). **c** Cutaway surface representation of CEI-bound NanoLuc tetramer. **d** Close-up views of CEI-binding pocket with residues creating the active site in stick representation. Key hydrogen bonds are shown as dashed yellow lines. **e** Superposition of FMA (yellow) and CEI (green) binding modes.

perfectly overlaps with the fatty acid-binding site, where decanoic acid molecules are found in the two previously determined NanoLuc complex structures, highlighting this site as a versatile ligand-binding pocket (Supplementary Fig. 3). Moreover, we found in our co-crystal structures that polyethylene glycol (PEG) molecules, originating from the mother liquor, bind in this pocket too (Supplementary Fig. 4).

From the superposition of our co-crystal structures, it is evident why neither CTZ nor its oxidized derivative CEI can bind to the luciferin-binding surface pocket in the same way as the one adopted by FMZ or its oxidized catalytic product FMA. As shown in Supplementary Fig. 5, the CEI molecule is indeed bulkier than the FMA, producing structural clashes when modeled in the FMA-preferred binding mode.

## NanoLuc is present as a monomeric protein at micromolar concentrations

We further investigated whether the NanoLuc homotetrameric association observed in the crystals might also exist in the solution. To test this hypothesis, we employed small-angle X-ray scattering (SAXS) analysis to probe the NanoLuc structure in solution. The SAXS profile of the micromolar solutions of NanoLuc closely fits the scattering profile calculated using a single NanoLuc monomer of the crystal structure ($\chi^2 = 2$), but consistently does not correspond at all to the scattering curve calculated using the dimer or tetramer (Supplementary Fig. 6a). In addition, an ab initio model reconstructed from the experimental SAXS data perfectly accommodates a monomeric form

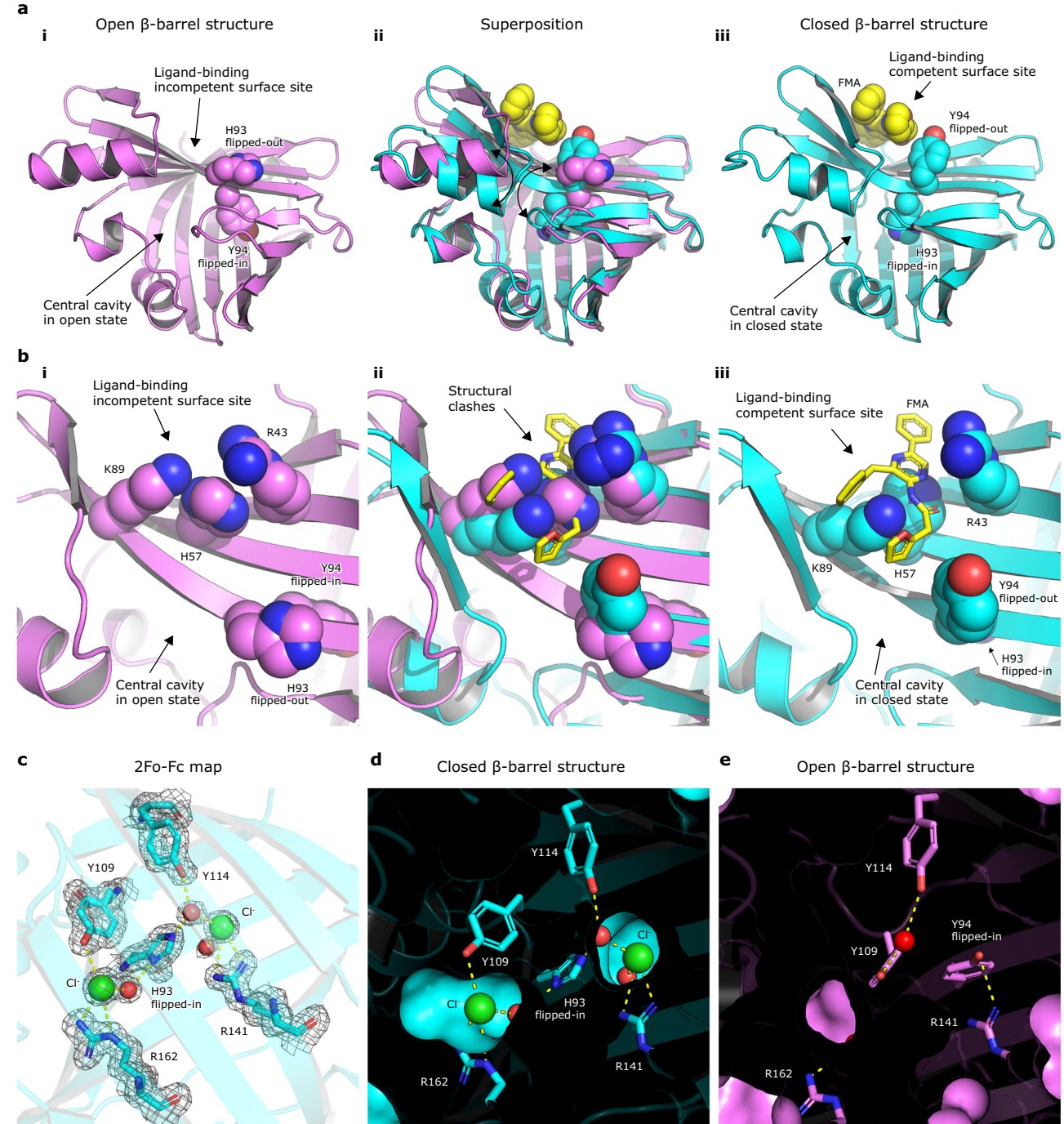

**Fig. 3 | Conformational switch between open and closed NanoLuc β-barrel structures. a** Cartoon representations of (i) ligand-free apo-NanoLuc structure (PDB ID: 5B0U), (iii) FMA-bound NanoLuc structure, and (ii) their superposition. The H93 and Y94 residues and FMA-luciferin are shown as space-filling spheres. **b** Close-up views of (i) luciferin-binding surface pocket in ligand-free apo-NanoLuc structure (PDB ID: 5B0U), (iii) FMA-bound NanoLuc structure, and (ii) their superposition. R43, H57, K89, H93 and Y94 residues are shown as space-filling spheres, and the FMA luciferin is shown as yellow sticks. **c** 2Fo-Fc electron density (contour level 1.2 σ) at the chloride-binding sites 1 and 2. **d**, **e** Cutaway surface representations of NanoLuc β-barrel interior in a closed (**d**) and open (**e**) state. Note that in the closed β-barrel state, the two chloride ions are bound inside the β-barrel.

of the NanoLuc luciferase (Supplementary Fig. 6b). Our SAXS results demonstrate that the NanoLuc luciferase in micromolar concentrations is indeed a monomeric enzyme in solution.

## A conformational switch between open and closed β-barrel structure

To better understand the role of an allosteric site shaped on the enzyme surface, we compared a ligand-free (apo-form) and ligand-

bound NanoLuc structures and revealed a conformational switch between the so-called open and closed states of the β-barrel structure (Fig. 3a, b). This conformational transition comprises several concerted structural re-arrangements, including unusual flipping of the β-strand S5. The open conformation of the β-barrel structure is captured in NanoLuc structures determined by Tomabechi[17] and Inouye[19]. One of the major hallmarks of this open conformation is that the side chain of H93 is exposed on the surface and concomitantly, the side

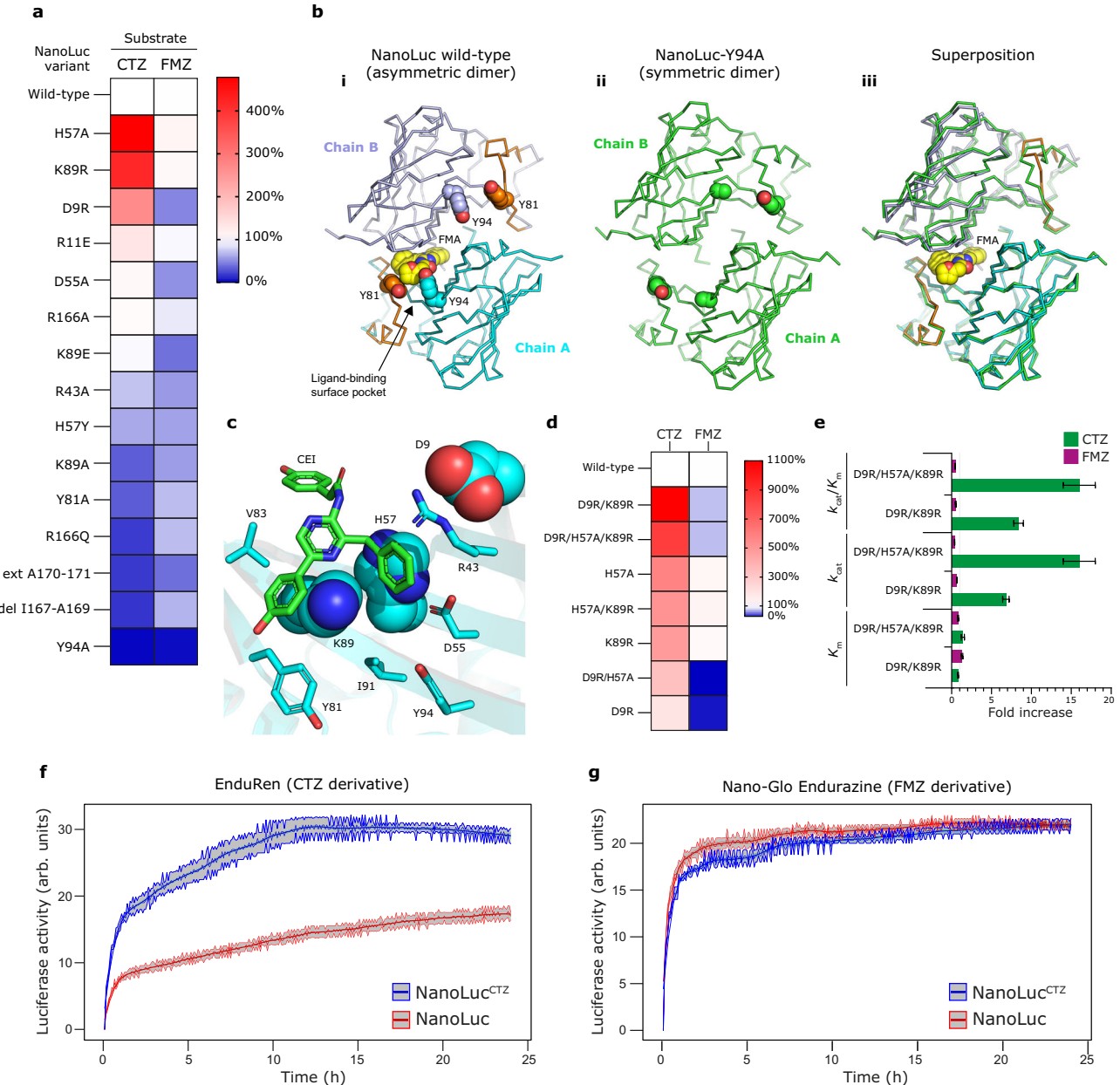

**Fig. 4 | Characterization of NanoLuc mutants. a** Heatmap showing the relative luciferase activities of NanoLuc mutants with CTZ (left column) and FMZ (right column) luciferins. NanoLuc wild-type = 100%. **b** Structures of FMA-bound asymmetric dimer of NanoLuc wild-type (i) and luciferin-free symmetric dimer of NanoLuc-Y94A mutant (ii), and their superposition (iii). The tyrosine-tyrosine gate Y81 and Y94 (A94) residues and FMA-luciferin are shown as space-filling spheres. **c** Close-up view of the CEI-bound luciferin-binding surface site of NanoLuc luciferase. Three residues, namely D9, H57 and K89, whose mutations increase biolumi-nescence with CTZ are shown as cyan space-filling spheres. The other protein residues are shown as cyan sticks, and the CEI-luciferin is shown as green sticks. **d** Heatmap showing the relative luciferase activities of NanoLuc mutants with CTZ

(left column) and FMZ (right column) luciferins; NanoLuc wild-type = 100%. **e** Relative fold increases ± standard errors (s.e.) of kinetic parameters ($K_{\mathrm{m}}$, $k_{\mathrm{cat}}$ and $k_{\mathrm{cat}}/K_{\mathrm{m}}$) observed in the double NanoLuc-D9R/K89R and triple NanoLuc-D9R/H57A/K89R mutants; NanoLuc wild-type = 1. Absolute values of kinetic parameters ($K_{\mathrm{m}}$, $k_{\mathrm{cat}}$ and $k_{\mathrm{cat}}/K_{\mathrm{m}}$) are summarized in Table 1. **f**, **g** Long-term live-cell biolumi-nescence imaging of cultured ARPE-19 cells expressing either original NanoLuc or engineered NanoLuc[CTZ] and powered by EnduRen (**f**) or Nano-Glo Endurazine (**g**) substrate. The luciferase activities upon addition of the corresponding luciferin were measured by LuminoCell device[53], integration time was 5 min. Each data point represents the mean enzymatic activity of three replicates. The standard error is represented as a shaded area around the mean curve.

chain of adjacent Y94 is dipped inside the β-barrel structure, making the central cavity more voluminous, presumably to accommodate bulky luciferin molecule in the enzyme-substrate Michaelis complex. On the contrary, the complex structures with bound fatty acid or luciferin molecules in the ligand-binding surface site show the closed conformation, with the side chain of Y94 placed on the surface while the side chain of H93 is inserted in the β-barrel structure, making the central cavity less voluminous (Supplementary Fig. 7). The most

striking feature is that the competences to bind luciferin either in the surface pocket or in the central putative catalytic site are mutually exclusive, implying a so-called homotropic negative allostery mechanism. This means that a luciferin molecule can be bound at one time either in the surface allosteric site (closed β-barrel structure) or in the intra-barrel catalytic site (open β-barrel structure), but never in both simultaneously (Fig. 3). Moreover, when the β-barrel adopts its closed state, two chloride ions can bind to the pre-formed intra-barrel

**Table 1 | Kinetic parameters of NanoLuc luciferase mutants**

| Enzyme variant | $K_m$/µM | $k_{cat}$/s$^{-1}$ | $K_p$/µM | $k_{cat}/K_m$/s$^{-1}$ µM$^{-1}$ |
|---|---|---|---|---|
| CTZ-luminescence | | | | |
| NanoLuc | 0.57 ± 0.02 | 2.48 ± 0.05 | 0.256 ± 0.005 | 4.4 ± 0.2 |
| NanoLuc-Y94A | 3.74 ± 0.09 | 0.225 ± 0.004 | 0.79 ± 0.04 | 0.060 ± 0.001 |
| NanoLuc-D9R/K89R | 0.46 ± 0.01 | 16.8 ± 0.6 | 0.163 ± 0.006 | 37 ± 1 |
| NanoLuc-D9R/H57A/K89R | 0.77 ± 0.07 | 40 ± 4 | 0.23 ± 0.02 | 70 ± 7 |
| FMZ-luminescence | | | | |
| NanoLuc | 0.123 ± 0.004 | 7.88 ± 0.03 | 0.56 ± 0.01 | 64 ± 2 |
| NanoLuc-Y94A | 1.29 ± 0.02 | 0.519 ± 0.007 | 0.85 ± 0.02 | 0.402 ± 0.006 |
| NanoLuc-D9R/K89R | 0.157 ± 0.009 | 5.2 ± 0.1 | 0.39 ± 0.03 | 33 ± 2 |
| NanoLuc-D9R/H57A/K89R | 0.098 ± 0.005 | 2.74 ± 0.03 | 0.34 ± 0.02 | 28 ± 1 |

Data are presented as mean with standard deviations (n = 3).

chloride-binding sites and thus inactivate the catalytic machinery (Fig. 3c–e). Our structures thus provide a structural basis for the reversible inhibition of NanoLuc luciferase by a high concentration of chloride ions, as previously reported by Altamash et al.[18].

**Restructuring the allosteric site boosts CTZ-bioluminescence**
Interestingly, 7 out of 16 amino acid residues mutated during the engineering of NanoLuc luciferase[8] (Fig. 1i) are found around the ligand-binding surface pocket (Supplementary Fig. 8). To probe the functional importance of the amino acid residues forming this surface pocket, we performed structure-guided mutagenesis and studied the effect of these mutations. All constructed mutants were expressed and purified as soluble proteins (Supplementary Fig. 9). As seen by gel filtration, it turned out that some single-point mutants, such as "reverse" mutations R11E and R43A, did not exist as pure monomeric proteins anymore but rather as monomer-tetramer mixtures (Supplementary Fig. 10). These mutations are located at the protein surface, mediating contacts in homotetramers observed in the crystals. Moreover, we observed strikingly different behavior of many mutants when either FMZ or CTZ was used in the reaction, confirming the assumption that these two luciferins can be differently recognized by the enzyme (Fig. 4a and Supplementary Table 2). For instance, truncation of three C-terminal residues (del 167–169) or two-alanine extension of the C-terminal end (ext A170–A171) substantially decreased CTZ-luminescence but had a moderate effect on FMZ-luminescence, demonstrating that CTZ reaction relies on intact C-terminal end. The most severe effect on both FMZ and CTZ luminescence was observed when a tyrosine-to-alanine mutation was introduced at position 94. The Y94A mutant severely compromised NanoLuc action through ~67- and ~130-fold reduction of catalytic efficiency ($k_{cat}/K_m$) with CTZ and FMZ, respectively (Table 1 and Supplementary Figs. 11–13). In addition, we obtained diffraction-quality crystals of the NanoLuc-Y94A mutant and solved its structure (Fig. 4b and Supplementary Table 1). Although this mutant was crystallized in identical conditions as the wild-type enzyme, it crystallized as a perfectly symmetric dimer with no bound luciferin molecule. As seen in our co-crystal structures, the side chain of Y94, together with the side chain of Y81, constitute a tyrosine-tyrosine dyad gating the ligand-binding surface pocket. This suggests a mechanism of allosteric interplay between the surface binding site and the intra-barrel catalytic site, which is compromised by the tyrosine-to-alanine mutation at position 94.

On the contrary, several mutations restructuring the allosteric site substantially and selectively increased bioluminescence with CTZ but not with FMZ (Fig. 4a and Supplementary Table 2). The most striking mutations, particularly D9R, H57A and K89R (Fig. 4c), individually induced up to ~4.5-fold enhancement of CTZ-bioluminescence. Subsequent combining of these mutations in corresponding double- and

triple-mutants resulted in up to a >10-fold increase in CTZ-bioluminescence, while FMZ-bioluminescence was slightly decreased (Fig. 4d). The two top-ranking mutants, namely the double-mutant D9R/K89R and the triple mutant D9R/H57A/K89R, the latter mutant termed as NanoLuc$^{CTZ}$, were selected for comprehensive kinetic characterization to better understand why these mutations improved selectively bioluminescence with CTZ but not with FMZ (Fig. 4e, Table 1 and Supplementary Figs. 14–16). The luminescence-emission spectra of engineered luciferase variants are red-shifted by ~2 nm with CTZ-luciferin, while not affected with FMZ-luciferin (Supplementary Fig. 17). The kinetic analyses showed a 15.9-fold increase of catalytic efficiency ($k_{cat}/K_m$) with CTZ-luciferin, which was achieved through a 16.1-fold increase in $k_{cat}$, while the $K_m$ and $K_p$ values were not significantly affected. Importantly, the triple mutant, NanoLuc$^{CTZ}$, displayed catalytic efficiency with CTZ ($k_{cat}/K_m = 70 \pm 7$ s$^{-1}$·µM$^{-1}$) which is very similar to that of the original NanoLuc with FMZ ($k_{cat}/K_m = 64 \pm 2$ s$^{-1}$·µM$^{-1}$).

**An engineered NanoLuc$^{CTZ}$ is a superior in vivo reporter**
We further engineered mammalian ARPE-19 cells to express either NanoLuc$^{CTZ}$ or NanoLuc luciferase for long-term live-cell imaging experiments. Notably, we demonstrated that the NanoLuc$^{CTZ}$ luciferase exhibits superior bioluminescence over the original NanoLuc in cell-based assays with EnduRen substrate (CTZ derivative designed for use in bioluminescence reporter assays), confirming its advantageous reporting properties (Fig. 4f and Supplementary Figs. 18 and 19). On the other hand, when the Nano-Glo Endurazine substrate (FMZ derivative designed for use in bioluminescence reporter assays) was used, the NanoLuc$^{CTZ}$ and NanoLuc showed similar luciferase activities, which however were substantially lower compared to NanoLuc$^{CTZ}$ activity with EnduRen substrate (Fig. 4g). Collectively, our cellular experiments showed that the engineered NanoLuc$^{CTZ}$ and EnduRen substrate represent a superior luciferase-luciferin reporting pair for long-term live-cell imaging applications, substantially surpassing the original NanoLuc/Nano-Glo Endurazine pair.

**Capturing azacoelenterazine bound in the intra-barrel catalytic site**
To obtain further structural insights into the NanoLuc$^{CTZ}$ catalysis, we attempted its co-crystallization with native CTZ or the non-oxidizable substrate analog azacoelenterazine (azaCTZ) we developed recently[21]. While no diffraction-quality crystals were obtained with CTZ, the co-crystallization with azaCTZ resulted in well-diffracting crystals, and the corresponding complex structure was determined (Supplementary Table 1). Inspection of the electron density map unambiguously revealed azaCTZ molecule bound to the catalytic cavity located inside its open state β-barrel structure (Fig. 5a–c). The azaCTZ triazolopyrazine core is placed in the center of the β-barrel structure, surrounded

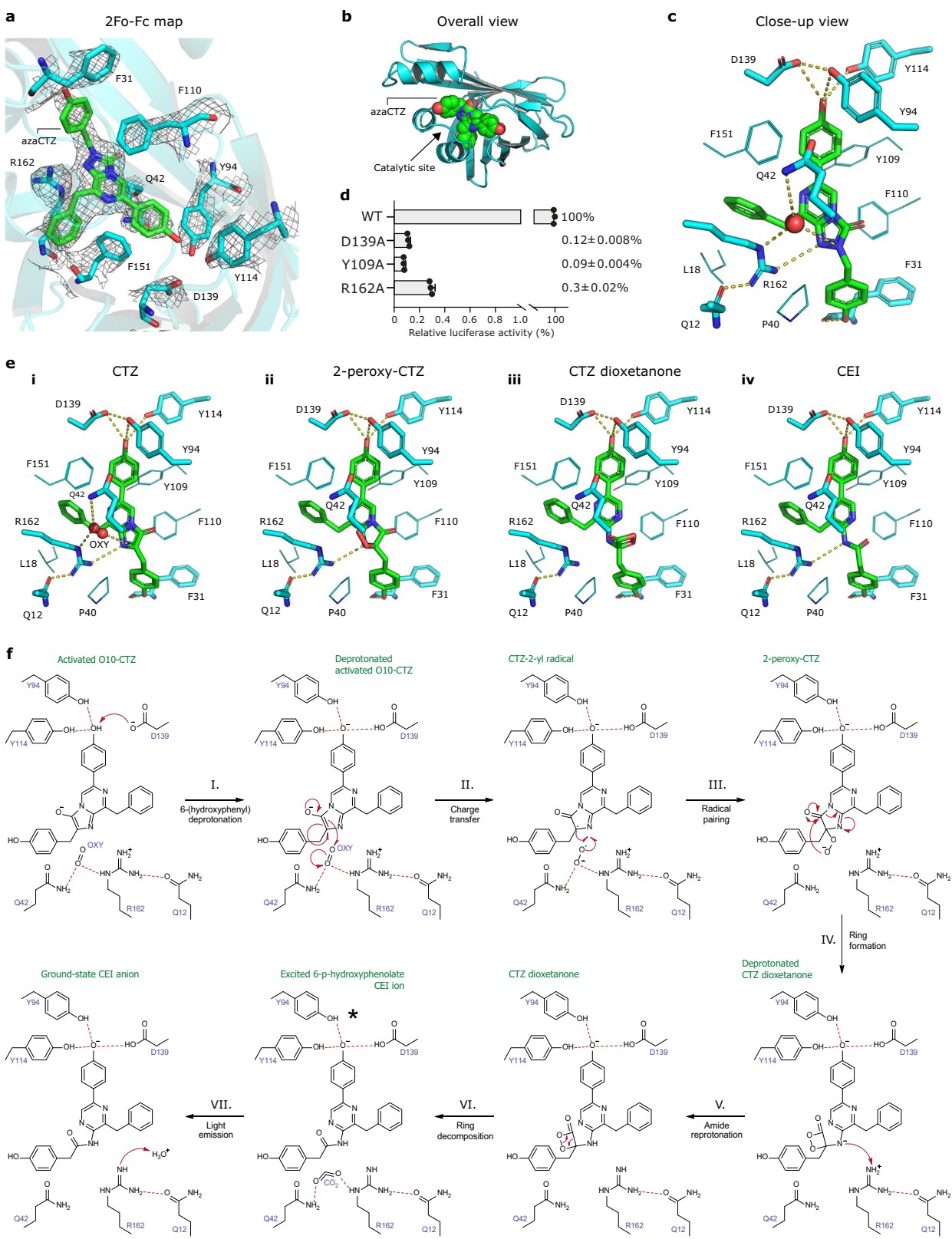

by multiple hydrophobic and aromatic residues, with the exception of a few polar residues. Precisely, the side chain of R162 is in close contact with the N1-nitrogen of azaCTZ (~3.5 Å), suggesting its crucial role in the protonation step of the CEI product. Moreover, the R162 interacts with a side chain of Q12 (~2.4 Å) on one side and with a water molecule found over the triazolopyrazine core (~3.3 Å) on the other side. The position of the water molecule may represent the positioning of a

dioxygen molecule prior to it attacking CTZ-luciferin at the C2-carbon atom during monooxygenation reaction. Mechanistically, this observation suggests a cardinal dual-function role for the R162 in both the positioning of luciferin and dioxygen molecules for their interaction and the protonating the CEI anion at amide group to secure high emission intensity. Notably, no residue that could potentially mediate initial deprotonation of the CTZ-luciferin at the N7-nitrogen or its

**Fig. 5 | Catalytic mechanism for NanoLuc-type luciferase reaction. a** 2Fo-Fc electron density map (contour level 3.0 σ) of azaCTZ bound in the intra-barrel catalytic site of NanoLuc[CTZ]. **b** Cartoon representation of the overall structure of NanoLuc[CTZ] (cyan) with bound azaCTZ (green). **c** Close-up view of azaCTZ (green) bound to NanoLuc[CTZ], with important residues creating the active site in cyan stick representation. Red sphere; a water molecule bound in the catalytic site. Key molecular contacts are shown as dashed yellow lines. **d** Mutagenesis of NanoLuc active site residues. Data indicate the average relative luciferase activities of each mutant. Assays were done in triplicate; error bars represent standard deviations. **e** Visualizations of the NanoLuc[CTZ] catalytic site with modeled CTZ (i), 2-peroxy-CTZ (ii), CTZ dioxetanone (iii), and CEI (iv). Key protein residues are shown as sticks and lines, molecular oxygen (OXY) is shown in red sphere, and hydrogen bonds are shown as dashed lines. **f** A blueprint for NanoLuc-type reaction mechanism. The cycle starts with the binding of CTZ into the catalytic site localized inside the β-barrel structure, and it enters with a deprotonated imidazopyrazinone core, as

demonstrated experimentally in previous work[21]. I. Upon the entry, the -OH group of the C6-(*p*-hydroxyphenyl) moiety is hydrogen-bonded with two tyrosines (Y94 and Y114), as well as deprotonated by D139, yielding the activated dianion O10-CTZ. Then, the side chain of R162, and perhaps helped by the side chain of Q42, position a co-substrate molecule (dioxygen) such that it can be attacked by the C2 carbon of O10-CTZ. II. The initial interaction proceeds via a charge-transfer radical mechanism. III. The next steps encompasses radical pairing and termination, resulting in the 2-peroxy-CTZ anion, which then undergoes intramolecular cyclization via a nucleophilic addition-elimination mechanism. IV. This cyclization generates a dioxetanone intermediate with a deprotonated amide group. V. The side chain of R162 protonates the amide group of oxyluciferin in order to avoid attenuation of the luminescence. VI. The energy-rich dioxetanone intermediate is unstable and decomposes by decarboxylation into an excited CEI product. VII. When returning to the ground state, the excited CEI releases a blue photon. Finally, the protonation status of R162 is restored by the proton transfer from a water molecule.

O10H tautomer is observed, which is similar to *Renilla*-type luciferases[21]. The catalytic relevance of R162 was then confirmed by mutagenesis experiment (Fig. 5d).

The R[2] 6-(*p*-hydroxyphenyl) substituent is deeply anchored in the β-barrel structure, where its hydroxyl group is simultaneously hydrogen bonded by side chains of D139 (~3.3 Å), Y114 (~3.6 Å) and Y94 (~3.6 Å). The tuning of the electronic state of CEI product and promoting the formation of the blue light-emitting phenolate anion is a common feature observed in structurally unrelated luciferases[21,23]. The remaining two substituents, R[1] 6-(*p*-hydroxyphenyl) and R[3] 8-benzyl, make predominantly hydrophobic and aromatic contacts with surrounding intra-barrel residues, with the exception of terminal hydroxyl moiety of R[1] 6-(*p*-hydroxyphenyl) substituent that forms a hydrogen bond with the backbone carbonyl of F31 (~2.6 Å). The functional importance of highlighted residues was verified by mutagenesis experiments (Fig. 5d).

## A proposal for the reaction mechanism of NanoLuc-type catalysis

The gained knowledge of the luciferase-luciferin recognition, supported with biochemical and biophysical experiments, allowed us to propose a reaction mechanism for the oxidative mechanism by which NanoLuc-type luciferases generate blue photons. We used azaCTZ-bound NanoLuc[CTZ] complex structure as a template to model the binding modes of native CTZ, the intermediates 2-peroxy-CTZ and CTZ dioxetanone, as well as the CEI oxyluciferin in NanoLuc active site (Fig. 5e). The proposed catalytic reaction mechanism is schematically depicted in Fig. 5f. First, the deprotonated CTZ enters the catalytic site inside the β-barrel structure. We recently demonstrated that the imidazopyrazinone core of CTZ is readily deprotonated in solution because its p$K_a$ of 7.55 is close to the physiological pH[21]. Upon luciferin entry, we deduce that the -OH group of the R[2] 6-(*p*-hydroxyphenyl) substituent is deprotonated by aspartate 139 to result in the dianionic O10-CTZ. The co-crystal azaCTZ-bound NanoLuc[CTZ] structure suggests that the side chain of R162 helps to position the co-substrate (O$_2$) such that it can be attacked by the C2 carbon atom of the activated luciferin. The initial interaction occurs via a charge-transfer mechanism, involving radical intermediates, which is analogous to a *Renilla*-type luciferase reaction[21]. The radical pairing and termination yields a 2-peroxy-CTZ anion, which then undergoes an intramolecular cyclization through a nucleophilic addition-elimination mechanism. As a result, a highly unstable energy-containing dioxetanone structure is formed. At this step, the deprotonated amide group of oxyluciferin is protonated by the side chain of R162. The dioxetanone ring is highly unstable and decomposes into an excited CEI product. When returning to the ground state, the excited CEI oxyluciferin emits a blue photon. The residue R162 is then reprotonated from a water molecule (Fig. 5f). Finally, we think that the conformational transition from open-to-

closed β-barrel structure (Fig. 3) may help to unload the CEI product and regenerate enzyme for the next catalytic cycle.

The reaction mechanism of NanoLuc luciferase proposed on our results is now supported by the recently deposited 3-methoxy-furimazine-bound NanoLuc structure (PDB ID: 7SNT). The binding mode of 3-methoxy-furimazine is very similar to the one we observed for azaCTZ, implying the analogous oxidative mechanism (Supplementary Fig. 20).

## Molecular docking confirms two luciferin-binding sites

To further validate crystallographic structures, we performed blind molecular docking where luciferins were docked into either a monomeric or a dimeric form of the closed β-barrel NanoLuc structure (PDB ID 8AQ6). The docking poses were compared with the crystallographic FMA and CEI-binding modes shown in Fig. 2e. The experiments with the monomeric NanoLuc have demonstrated that the luciferin molecules tend to bind in the entrance vestibule toward the central cavity, but none of the FMZ or CTZ poses were bound inside the catalytic pocket nor the allosteric site identified in our co-crystal structures (Supplementary Fig. 21). When the docking was constrained to the allosteric site, the binding energy was almost 2 kcal/mol worse, showing a higher affinity of luciferins toward the position at the β-barrel entrance vestibule above the H2 and H3 helices.

By contrast, when the homodimer NanoLuc structure was used as the receptor model, the docking results showed that there are two energetically equivalent luciferin-binding sites; the first site is in the β-barrel entrance vestibule also found in monomeric NanoLuc, and the second pose is in the surface-located allosteric site (Fig. 6a and Supplementary Fig. 22). The molecular docking thus suggests that luciferins tend to bind to the intra-barrel catalytic site but also to the allosteric site. However, the binding to the latter surface site would require the conformational transition into the closed β-barrel structure. The docking results are summarized in Supplementary Table 3.

## Tracking the route of luciferin into the catalytic site

We then attempted docking of FMZ into the active site of different NanoLuc structures, with the closed β-barrel (PDB IDs 8AQ6, 5IBO, and 7MJB) and with the open β-barrel structures (PDB IDs: 5BOU, 8BO9, and 7VSX). The predicted binding energies differed significantly among the different structures (Supplementary Table 4). The docking results suggest that luciferin binding inside the intra-barrel active site of NanoLuc is favorable only in the open β-barrel conformation. At the same time, it is unlikely the luciferin could bind inside of NanoLuc with the closed β-barrel structure. Notably, the binding mode of azaCTZ in the NanoLuc[CTZ] crystal complex (Fig. 5a) was reproduced precisely by the docking calculations for both FMZ and CTZ luciferins with the high affinities of −11.6 and −12.2 kcal/mol, respectively (Supplementary Fig. 23). Accordingly, theses docking experiments suggest that both

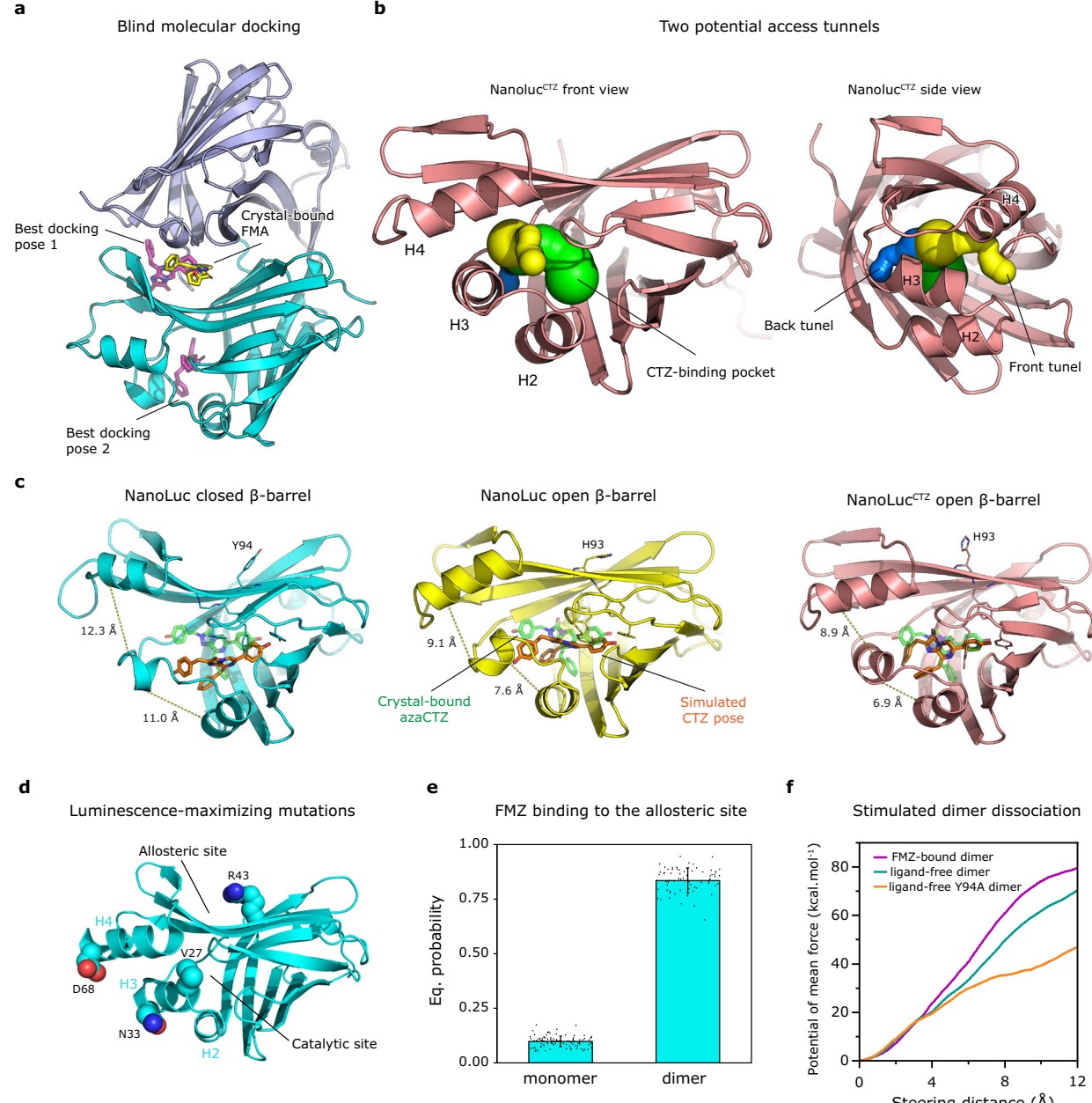

**Fig. 6 | Molecular docking and computational simulations of NanoLuc complexes. a** Visualization of the best poses of FMZ (magenta sticks) from molecular docking to the NanoLuc monomer dimer (chain A cyan and chain B blue cartoon), in comparison with the crystal-bound FMA (yellow sticks). **b** Two potential access tunnels (yellow and blue spheres) to the central luciferin binding cavity (green spheres) of NanoLucCTZ (salmon cartoon) are shown from the front and the side view. The tunnels were computed on Caver Web[84]. **c** The final poses of CTZ (orange sticks) from ASMD simulations of NanoLuc structures in the closed and open β-barrel state, and NanoLucCTZ open β-barrel structure. The crystal-bound azaCTZ (transparent green sticks) is shown in each picture for reference. The interacting aromatic residues are shown as thin sticks. The distances (in Å) between H2/H3 and H3/H4 helices are indicated by yellow dashed lines. Associated MD trajectories are shown in Supplementary Movies 1–3. **d** NanoLuc closed β-barrel structure with shown mutations (L27V, K33N, K43R, and Y68D) introduced during the last third phase of NanoLuc development responsible for maximizing FMZ fluorescence. **e** Equilibrium probabilities of FMZ macrostates bound in the surface pocket of monomeric and dimeric NanoLuc. The calculation was bootstrapped 100 times using a random 80% of the data. The height of the bars corresponds to the mean value and the error bars show the standard deviation. **f** The potential of mean force needed to steer the two subunits of different NanoLuc dimers up to 12 Å apart, obtained by adaptive steered molecular dynamics.

CTZ-to-CEI and FMZ-to-FMA reactions proceed through identical conversion mechanism, as depicted in Fig. 5f.

Moreover, we applied adaptive steered molecular dynamics (ASMD) to probe the luciferin-binding pathway toward the intra-barrel catalytic site. We identified two hypothetical access tunnels, the first one localized between H2 and H3 helices (back tunnel), and the second one (front tunnel) between H3 and H4 helices (Fig. 6b). Being pulled through the first tunnel, the simulated CTZ-luciferin ended in a similar

position as the azaCTZ co-crystallized with NanoLucCTZ (Supplementary Movies 1–3), which was not achieved using the second tunnel. Furthermore, the luciferin was predicted to bind at the mouth of the first tunnel by molecular docking (Fig. 6a); therefore, we deduce that the H2/H3 tunnel might be more relevant.

Conformations of CTZ bound to the three simulated NanoLuc structures were compared to the crystal-bound conformation seen for the NanoLucCTZ structure (Fig. 6c). In the NanoLucCTZ luciferase, the

conformation was well reproduced, while in the other structures, the simulated poses of CTZ deviated from the crystal binding mode. The most prominent difference between the simulated and crystal poses is in the closed β-barrel structure, where the simulated CTZ is shifted by ~4 Å from the crystallographic mode, toward the H2 helix. This shift is likely caused by the aromatic interaction of the luciferin core with the side chain of either Y99 or F100, as observed in different replicas of the MD simulations. Moreover, unlike the open β-barrel structures, the closed NanoLuc became looser during the simulation (the distance between H2/H3 and H3/H4 increased by over 3 Å each) to enable the luciferin entry into the catalytic pocket. Interestingly, in the last phase of the NanoLuc development, which maximized the luminescence with FMZ, four mutations (L27V, K33N, K43R, and Y68D) were introduced (Fig. 6d)[8]. While the R43 points toward the surface allosteric site, the other three residues are located in the region of H2, H3, and H4 helices, highlighting the functional importance of these dynamic protein elements.

## A luciferin-triggered stabilizing effect

Next, we analyzed the behavior of FMZ in monomeric and dimeric NanoLuc structures using adaptive sampling simulations with the root-mean-square deviations (RMSDs) of protein Cα-atoms as the adaptive metric, in a total simulation time of 10 µs. Markov state models (MSMs) were constructed using the RMSD of FMZ as the metric to cluster the simulations. The implied timescales plots and Chapman–Kolmogorov tests show the two transitions, which hints toward three or more macrostates (Supplementary Fig. 24). Individual states were used to calculate kinetics and binding affinity of FMZ to the monomeric and dimeric NanoLuc. For both systems, three macrostates were constructed (Supplementary Fig. 25). In the monomeric form, only one macrostate described the FMZ bound in the surface allosteric site, with

only about 10% probability (Fig. 6e). Most of the time, FMZ was bound near the H4 helix or interacted with other parts of the protein. On the other hand, in the enzyme dimeric form, two macrostates showed FMZ localized in the surface allosteric site, with more than 83% probability in total. Interestingly, FMZ had no tendency to bind into the β-barrel interior, indicating that any rearrangement of the binding site necessary for binding is not induced by the presence of the ligand near the β-barrel entrance. Finally, the kinetics and thermodynamics of the FMZ unbinding were calculated from the MSMs (Supplementary Table 5). Overall, the negative ΔG, high equilibrium probability (Fig. 6e), and lower FMZ RMSD distribution (Supplementary Fig. 26) indicated the strong preference of FMZ to bind to the surface allosteric site in the enzyme dimeric form, but not when it is monomeric.

The effect of the FMZ binding on the NanoLuc dimer was then analyzed by comparing adaptive sampling simulations of this dimer in the presence and absence of FMZ-luciferin. MSMs were constructed based on the RMSD of the protein Cα atoms to track the associative states of the two monomeric units, which resulted in three and four macrostates, respectively (Supplementary Figs. 27–29). The affinity and the equilibrium probability of the dimer, when complexed with ligand, show a strong stabilizing effect of FMZ on the dimeric form of NanoLuc luciferase (Supplementary Table 6 and Supplementary Fig. 30). In addition, the ASMD method was employed to simulate the dissociation of NanoLuc dimer with and without bound FMZ-luciferin (Fig. 6f). The potential of the mean force needed to dissociate the dimer complexed with the bound luciferin was about 10 kcal/mol higher compared to the ligand-free dimer, which implies a stabilizing effect of FMZ-luciferin on the enzyme dimer. This behavior resembles the mechanism of so-called molecular glue molecules that mediate protein-protein interactions of normally monomeric proteins[24, 25]. In addition, the X-ray structure of the NanoLuc carrying Y94A mutation,

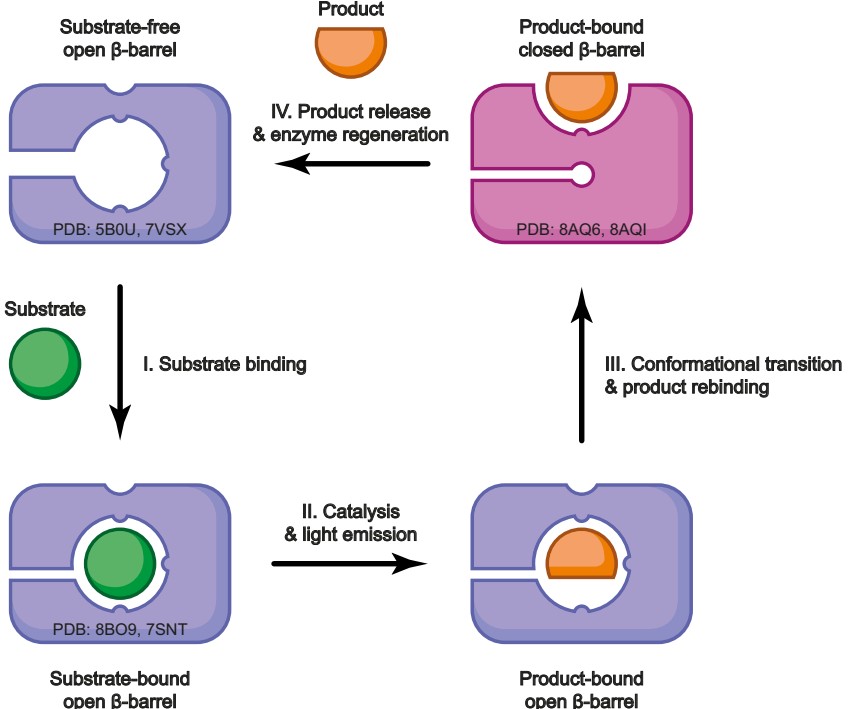

**Catalytic cycle of NanoLuc luciferase**

**Fig. 7 | Structure-based model for NanoLuc luciferase action.** I. A substrate molecule enters the intra-barrel catalytic site, the luciferase retains an "open β-barrel" conformation. II. Catalytic conversion of the substrate into its reaction product, followed by the emission of a blue photon. III. Release of the product out of the intra-barrel catalytic site. This step is accompanied by an open-to-closed conformational transition and a rebinding of the product to the newly formed allosteric site on the protein surface. IV. Dissociation of the product from the surface allosteric site, allowing recycling into the pre-catalytic "open β-barrel" state. The PDB ID codes for representative crystallographic structures are provided.

which severely compromised the luciferase activity (Fig. 4), was analyzed by ASMD too (Fig. 6f). The potential of mean force in the NanoLuc-Y94A mutant was decreased by ~25 kcal/mol compared to the dimer formed by the wild-type enzyme. The low affinity of the dimer in this mutant highlights the functionally important role of the Y94 in the NanoLuc function.

## A structure-based model for NanoLuc luciferase action

By using X-ray crystallography, we could visualize in this work some luciferin-bound states of NanoLuc luciferase, and thus secure additional insights into the puzzle of its molecular bioluminescence mechanism. From these, along with previously captured crystallographic snapshots as well as available biochemical and computational data, we are providing a structure-based model for this luciferase catalysis (Fig. 7). The most striking feature of this model is based on a conformational transition between the so-called "open β-barrel" and "closed β-barrel" structures. Our crystallographic data reveal that the luciferin substrate can bind to an intra-barrel catalytic site ("open β-barrel") as well as to a secondary allosteric site situated on the surface of the "closed β-barrel" conformation. Moreover, luciferin binding to the allosteric site will prevent its binding to the catalytic site, and vice versa, thanks to this conformational transition.

Our model assumes that when the substrate (luciferin) enters the intra-barrel catalytic site (Fig. 7i), it is catalytically converted into the product (oxyluciferin) and this is followed by the emission of a blue photon (Fig. 7ii). The structure-based mutagenesis experiments described in this work actually confirmed this assumption. Following this, the release of the reaction product from the intra-barrel catalytic site may be facilitated by an open-to-closed conformational transition (Fig. 7iii). At this stage, the reaction product may actually shift to the allosteric site appearing on the protein surface upon this conformational change. Our structural and kinetic data indicate that the native oxyluciferin (CEI) uses a radically different binding mode than the synthetic one (FMA), involving more extensive interactions with the allosteric site. This could explain why our structure-based restructuring of the allosteric site substantially boosted the CTZ-based luminescence but had no effect on the FMZ-based luminescence. Our kinetic measurements showed that NanoLuc efficiently binds FMZ ($K_m = 0.123 \pm 0.004\,\mu M$) with a minimal product inhibition ($K_p = 0.56 \pm 0.01\,\mu M$), whereas a less effective binding ($K_m = 0.57 \pm 0.02\,\mu M$) and substantial product inhibition ($K_p = 0.256 \pm 0.005\,\mu M$) is observed with native CTZ-luciferin. We suggest that the oxyluciferin product binding to the surface luciferin-binding site provides an allosteric-based negative feedback loop, restraining the β-barrel opening and halting the reaction after several cycles. This would lead to flash-type bioluminescence, which, unlike bacterial and fungal bioluminescence, is rather typical of marine one. Therefore, we think that the combination of the engineered NanoLuc luciferase with the synthetic FMZ-luciferin reported by Hall et al.[8] actually removed this allosteric-based negative feedback leading to a so-far unmatched glow-type bioluminescence. Our model also assumes that upon the ligand release from the allosteric site, the free enzyme will regenerate via a closed-to-open β-barrel transition (Fig. 7iv).

## Discussion

Bioluminescent biosensors have a wide range of practical applications and bioassay formats. For instance, in the present COVID-19 pandemic, bioluminescent technologies play a key role in the study of the causative virus SARS-CoV-2 as well as in the development of diagnostic assays and drug development via ultrahigh-throughput screenings[26–28]. NanoLuc is the brightest known luciferase. Its light-producing activity has been used across a huge application range[9–11,13,15]. However, the molecular basis underlying its catalysis remains puzzling[8,12,17,19], and this is mainly due to the lack of structural knowledge on catalytically favored luciferin-bound enzyme complexes.

To fill this gap, we attempted to capture luciferin-bound NanoLuc complexes through crystallographic experiments. The NanoLuc structure is formed of a 10-stranded β-barrel capped by 4 α-helices, exhibiting a structural similarity with non-catalytic fatty acid-binding proteins (FABPs), pointing toward their common evolutionary history[17]. A central intramolecular cavity is dominated by hydrophobic residues, although a few polar/charged residues are also present. Therefore, it has been postulated that the luciferin should primarily bind to the central cavity inside the β-barrel structure[8,17,19], where fatty acids are typically bound in related FABPs[29–31]. Despite this expectation, the crystal structures of NanoLuc complexed with decanoic acid (PDB IDs: 7MJB and 5IBO) showed that the fatty acid is not bound inside the β-barrel structure, but instead, it occupies a pocket shaped on the protein surface. Specifically, the decanoic acid wraps around a protrusion formed by the side chain of K89. Interestingly, our initial crystallization experiments also captured luciferin molecules bound to the same surface-localized pocket, reinforcing it as a luciferin-binding allosteric site.

A question that arises is why both fatty acid and luciferin molecules tend to bind to this surface allosteric site, and not only to the central cavity endowed with catalytic function, as captured in the azaCTZ-bound NanoLuc$^{CTZ}$ complex determined in this study. An explanation can be provided by the conformational transition between the so-called open and closed states of the β-barrel structure, which comprises several concerted structural re-arrangements, including unusual flipping of the S5 β-strand. Previously, the open conformation of the β-barrel structure was captured in Tomabechi[17] and Inouye[19] NanoLuc structures; the latter one contains 2-(N-morpholino)ethanesulfonic acid (MES) bound inside the β-barrel. One of the major hallmarks of this open conformation is that the side chain of H93 is exposed on the surface and concomitantly the side chain of Y94, identified as a critical residue for efficient catalysis, is dipped inside the β-barrel structure, making the central cavity more voluminous. The remaining ligand-bound NanoLuc structures determined so far show the closed conformation, with the side chain of Y94 placed on the surface while the side chain of H93 is inserted in the β-barrel structure. We anticipate that the protein motions accompanying this conformational transition between the open and closed states of the β-barrel structure can be important for efficient ligand binding/unbinding events. This behavior could be somehow reminiscent to the family of small non-specific lipid transfer proteins (nsLTPs)[32,33]. The nsLTPs also exhibit a central hydrophobic cavity, which is the primary lipid-binding site. However, structural studies showed that fatty acids are bound not only in the central cavity but also on the protein surface[32]. Madni et al. demonstrated lipid-induced conformational changes leading to the opening of the central cavity, representing a sophisticated gating mechanism for the entry and exit of transported fatty acids[32,33]. Analogously, it is apparent that an energetic barrier of conformational transition between the open and closed states of the β-barrel structure is high, and we speculate that luciferin molecules themselves could play a role in this gating process during NanoLuc action. Mechanically, the ligand binding to the allosteric site prevents the simultaneous binding to the catalytic site, and vice versa, through concerted conformational changes, revealing the so-called homotropic negative allostery mechanism.

Another issue that should be considered carefully is the crystal packing. The NanoLuc is reported to be a monomeric enzyme[8,12], but all its crystal structures captured in the closed state of β-barrel structure display tight homotetrameric association. Our SAXS experiments proved that the NanoLuc indeed exists as a monomeric protein in micromolar concentrations. The closed state of the β-barrel structure seems to prevent the ligand binding to the intra-barrel catalytic site, but at the same time allows the binding of a ligand (e.g., fatty acid or luciferin molecule) into the allosteric site. Structurally, the complexation of crystallographic NanoLuc homodimers/homotetramers having

ligand bound in the surface allosteric site resembles the action mechanism of so-called molecular glue (MG) molecules[24,25]. The MG-systems are characterized by the lack of ligand binding in at least one protein partner and an under-appreciated pre-existing low micromolar affinity between the two proteinaceous subunits that is enhanced by the ligand to reach the nanomolar range[24]. We hypothesize that the enzyme tetramerization captured in the crystals may reflect the inherent feature of native *Oplophorus* luciferase although it would have been mostly suppressed in the engineered NanoLuc. The construction and analysis of several NanoLuc "reverse" mutants in this work support this hypothesis. It turned out that some single-point mutants, such as R11E and R43A, did not exist as pure monomeric proteins anymore but rather as monomer-tetramer mixtures. We therefore speculate that in "real life", such quaternary structures could serve as an "inactive" luciferin-loaded storage form of *Oplophorus* luciferase. Indeed, MD simulations performed in this work imply that the presence of luciferin molecule bound in the surface allosteric site has a positive effect on the enzyme self-association when it adopts the closed β-barrel structure. Therefore, we speculate that such inactive enzyme-luciferin complexes could function as a storage pool of luminescent agents in *O. gracilirostris*, representing an elegant evolutionary solution to obviate the need to encode an extra luciferin-protecting protein, as described in other marine luminescent organisms[34,35]. Some "reverse" luciferase mutants generated in this work did not exist as a pure monomeric protein anymore but rather as monomer-tetramer mixtures, supporting this hypothesis. Moreover, we identified two chloride-binding sites inside the β-barrel structure, and several additional chloride-binding sites on the enzyme surface, suggesting that chloride ions may contribute to the NanoLuc inactivation through a closing of its β-barrel structure. Recently, Altamash et al. reported that elevated concentrations of chloride ions reversibly inactivated NanoLuc[18], confirming this assumption. Furthermore, we previously showed that *Renilla*-type luciferases are also inhibited by halide ions[21], highlighting the convergent evolution of a regulatory mechanism of marine luciferases.

It was unclear whether the binding of luciferin molecules to the surface allosteric site could be a crystal lattice-biased artifact or not, and whether it has any catalytically important role. To address this concern, we performed comprehensive mutagenesis of the luciferin-binding surface pocket of NanoLuc highlighting its functional importance. The mutagenesis experiments showed substantial functional impairments when key luciferin-interacting residues were mutated, identifying Y94 as the most important residue for both FMZ and CTZ bioluminescence. Surprisingly, some mutations selectively boosted bioluminescence with CTZ-luciferin but not with FMZ-luciferin. The combined triple (D9R/H57A/K89R) mutant, termed NanoLuc$^{CTZ}$, exhibited superior catalytic properties with widely accessible CTZ-luciferin ($k_{cat}/K_m = 70 \pm 7$ s$^{-1}$·µM$^{-1}$), surpassing the level of catalytic efficiency of original NanoLuc/FMZ pair ($k_{cat}/K_m = 64 \pm 2$ s$^{-1}$·µM$^{-1}$). Indeed, in vivo experiments confirmed superior CTZ-luminescence of NanoLuc$^{CTZ}$ in mammalian cells, highlighting its suitability for use in optical assays and biosensors. Selective enhancement of bioluminescence can be explained by radically different binding modes of CTZ and FMZ in the surface allosteric pocket, as observed in our co-crystal structures. Our experiments thus evidenced the existence of a communication pathway between the allosteric and catalytic sites. Notably, we showed that restructuration of the allosteric site can dramatically enhance catalysis in the remote active site localized inside the β-barrel structure.

Recently, we showed that CTZ-to-CEI conversion proceeds via a charge-transfer radical mechanism[21]. The azaCTZ-bound NanoLuc$^{CTZ}$ complex structure determined in this work revealed the nature of the catalytic machinery, enabling us to propose the reaction mechanism for the NanoLuc-type reaction. Mechanistically, an intra-barrel R162 navigates the imidazopyrazinone component of luciferin to attack O$_2$ via a radical charge-transfer mechanism, as well as it protonates the excited amide product to secure high emission intensity. Previous works evidenced that the phenolate anion is the blue emitter in CTZ-bioluminescence[21,23,36–38]. For example, in *Renilla*-type luciferases, the phenolate anion of the CEI product is generated via deprotonation by an aspartate residue[21]. Surprisingly, a similar constellation is observed in NanoLuc, where an aspartate (D139), supported by two tyrosines (Y94 and Y114), also fine-tunes the electronic state of amide product, promoting the formation of the phenolate anion that emits blue photons. The aspartate-mediated deprotonation of oxyluciferin appears as a common feature employed by CTZ-utilizing marine luciferases[21]. Altogether, knowledge obtained in this study will contribute to the understanding of NanoLuc's puzzling mechanism, and can be exploited for the rational design of luciferase–luciferin pairs applicable in ultrasensitive bioassays and/or bio-inspired light-emitting technologies.

## Methods

### Luciferins and aza-luciferin analogs syntheses
The concentrated solutions of CTZ and FMZ used in this work where obtained by the hydrolysis of, respectively, hikarazines-001 and hikarazines-086 as previously described[39]. Azacoelenterazine (azaCTZ) was synthesized as described previously[21]. Synthesis of azaFMZ is described in Supplementary Note 1.

### Mutagenesis and DNA cloning
Megaprimer PCR-based mutagenesis[40] was applied to create single-point mutations as well as for gene truncation and extension. The megaprimers with the desired mutation, truncation or extension were synthesized in the first PCR reaction using a mutagenic primer and one universal primer (Supplementary Table 7). The megaprimer was gel-purified purified by DNA electrophoresis and used as a primer in the second round of PCR to generate the complete DNA sequence with the desired mutation. After the PCR reaction, the original DNA template was removed by *Dpn*I treatment (2 h at 37 °C), and the mutated plasmid was transformed into chemically competent *Escherichia coli* DH5α cells. Plasmids were isolated from three randomly selected colonies and error-free DNA genes were confirmed by DNA sequencing (Euro-fins Genomics, Germany). Protein sequences of all NanoLuc variants generated and used in this work are aligned in Supplementary Fig. 31.

### Overexpression and purification of NanoLuc luciferases
*E. coli* BL21(DE3) cells (NEB, USA) were transformed with pET-21b plasmid encoding for N-terminally His-tagged NanoLuc gene, plated on LB-agar plates with ampicillin (100 µg/mL) and grown overnight at 37 °C. A few colonies were transferred and used to inoculate an aliquot of 100 mL of 2 × LB medium containing 100 µg/mL ampicillin followed by a 5-h incubation at 37 °C. The expression of NanoLuc was induced by the addition of IPTG to a final concentration of 0.5 mM. After overnight cultivation at 20 °C, the cells were harvested by centrifugation (15 min, 4000 × g, 4 °C) and resuspended in a TBS buffer A (10 mM Tris-HCl, 50 mM NaCl, 20 mM imidazole, pH 7.5) containing DNase (20 µg/mL). The cells were then disrupted by sonification using Sonic Dismembrator Model 705 (Fisher Scientific, USA). The lysate was clarified by centrifugation (50 min, 21,000 × g, 4 °C) using a Sigma 6-16K centrifuge (SciQuip, UK). The filtrated supernatant containing His-tagged NanoLuc was applied to a 5-mL Ni-NTA Superflow Cartridge (Qiagen, Germany) pre-equilibrated with TBS buffer A. NanoLuc was eluted with TBS buffer B (10 mM Tris-HCl, 50 mM NaCl pH, 250 mM imidazole, pH 7.5). Finally, NanoLuc was purified by size exclusion chromatography using Äkta Pure system (Cytiva, USA) equipped with HiLoad 16/600 Superdex 200 pg or Superdex 75 Increase 10/300 GL column equilibrated with a gel filtration buffer (50 mM NaCl, 10 mM Tris-HCl, pH 7.5). The protein purity was verified by SDS-PAGE. The same protocol was used for all NanoLuc variants.

## Crystallization experiments

Purified NanoLuc was concentrated to a final concentration of ~10 mg/mL using Centrifugal Filter Units Amicon[R] Ultra-15 Ultracel[R]−3K (Merck Millipore Ltd., Ireland). Concentrated NanoLuc was mixed with a 4 molar excess of FMZ non-oxidizable analog azaFMZ (stock solution of azaFMZ was 10 mM in isopropanol). The precipitated material was removed by centrifugation (10 min, 12,000 g, 4 °C) after 60 min incubation at 4 °C, and the supernatant was directly crystallized. The crystallization was performed in Easy-Xtal 15-well crystallization plates by a hanging drop vapor diffusion, where 1 μL of NanoLuc-azaFMZ mixture was mixed with the reservoir solution (200 mM magnesium chloride, 100 mM potassium chloride, 25 mM sodium acetate pH 4.0, and 33% PEG 400) in the ratio 1:1 and equilibrated against 500 μL of the reservoir solution. The crystals usually grew in 3–5 days. The good-looking crystals were soaked overnight in the mother liquor supplemented with 10 mM FMZ or CTZ, flash-frozen in the reservoir solution supplemented with 20% glycerol by liquid nitrogen and stored for X-ray diffraction experiments. The same crystallization protocol was used for the NanoLuc-Y94A mutant.

For the NanoLuc[CTZ] co-crystallization, the purified luciferase was concentrated to a final concentration of ~10 mg/mL and mixed with a 4 molar excess of azaCTZ luciferin[21]. The crystallization was performed in Easy-Xtal 15-well crystallization plates by a hanging drop vapor diffusion, where 1 μL of NanoLuc[CTZ]-azaCTZ mixture was mixed with the reservoir solution (100 mM ammonium acetate, 0.1 M Bis-Tris pH 5.5 mM and 17% (w/v) PEG 10000) in the ratio 1:1 and equilibrated against 500 μL of the reservoir solution. Crystals were observed at 20 °C after 5–7 days. Crystals were flash-frozen in the reservoir solution supplemented with 20% glycerol by liquid nitrogen and stored for X-ray diffraction experiments. No further optimization was necessary.

## Diffraction data processing and structure determinations

Diffraction data were collected at the Swiss Light Source synchrotron at the wavelength of 1.0 Å. The data were indexed and processed using XDS (version January 31, 2020)[41], and Aimless (implemented in CCP4 7.0.073) was used for data reduction and merging[42]. The initial phases of NanoLuc were solved by molecular replacement using Phaser[43] implemented in Phenix 1.19.2-4158[44]. The structure of NanoKaz (PDB ID: 5B0U)[17] was employed as a search model for molecular replacement. Twinning was detected by phenix.xtriage[44], and taken into account during reciprocal-space refinement steps using Refmac5[45]. For the NanoLuc/FMA complex, Refmac5 refined with four twin domains and twinning operators 0.606 (h, k, l), 0.214 (h, -k, -l), 0.089 (k, h, -l) and 0.09 (-k, -h, -l). For the NanoLuc/CEI complex, there were four twin domains and twinning operators were 0.485 (h, k, l) 0.166 (-h, -k, l), 0.180 (-k, -h, -l) and 0.169 (k, h, -l). The refinement was carried out in several cycles of automated refinement in Refmac5[45] and/or phenix.refine tool[44] and manual model building performed in Coot 0.8.9.2[46]. The chemical structures and geometry restraints libraries of FMA, CEI and azaCTZ were created using Ligand Builder and Optimization Workbench (eLBOW) implemented in Phenix 1.19.2-4158[44]. The final models were validated using tools provided in Coot 0.8.9.2[46]. Structural data were graphically visualized with PyMOL 1.8.4 Molecular Graphics System (Schrödinger, LLC). Atomic coordinates and structure factors were deposited in the Protein Data Bank[20] under the ID codes: 8AQ6, 8AQI, 8AQH and 8BO9 (Supplementary Table 1).

## Small-angle X-ray scattering (SAXS)

The SAXS datasets were collected using the BioSAXS-2000, Rigaku at CEITEC (Brno, Czech Republic). Data were collected at 293 K with a focused (Confocal Max-Flux, Rigaku) Cu Kα X-ray (1.54 Å). The sample to the detector (HyPix-3000, Rigaku) distance was 0.48 m covering a scattering vector ($q = 4\pi \sin(\theta)/\lambda$) range from 0.008 to 0.6 Å$^{-1}$. Size exclusion buffer (10 mM Tris-HCl, 50 mM NaCl, pH 7.5) was used for the blank measurement. NanoLuc samples were measured at a

concentration of 387 μM (3.0 mg.mL$^{-1}$) without luciferin or in the presence of luciferin (azaCTZ) at a NanoLuc:azaCTZ ratio of 1:4. Evaluation of solution scattering and fitting to experimental scattering curves was carried out using CRYSOL[47]. The structural models of NanoLuc monomer, dimer, and tetramer were created from the obtained crystal structure in PyMOL 1.8.4. Refined ab initio models were produced by DAMMIN, where the starting structure was generated by DAMAVER using 10 individual ab initio models produced by DAMMIF[48]. Superimposition of the atomic and ab initio models was carried out using PyMOL 1.8.4.

## Preparation of luciferin stock solutions for luminescence-measurements

Stock solutions of FMZ (Aobious, USA) and CTZ (Carl Roth, Germany) were prepared by dissolving a weighed amount of solid FMZ or CTZ in ice-cold absolute ethanol to obtain a 500 μM luciferin concentration. The stock solutions were stored in glass vials under a nitrogen atmosphere. The concentration and quality of the luciferin stock solutions were verified spectrophotometrically before each measurement.

## Measurement of specific luciferase activity

Specific luciferase activity of NanoLuc and its mutants was determined at 37 °C using a FLUOStar Omega microplate reader (BMG Labtech, Germany). Buffered FMZ and CTZ solutions were prepared by dilution of their ethanolic stock solution into 100 mM potassium phosphate buffer (pH 7.50) to obtain a 2.2 μM concentration of luciferin. Samples of 25 μL of purified enzyme solution were placed in microplate wells. After 10 s baseline collection, the luciferase reaction was initiated by injection of 225 μL of 2.2 μM buffered FMZ or CTZ solution and monitored for total luminescence (240–740 nm) for 15 s. The final enzyme concentration varied between 0.03 and 320 nM and was tailored to each enzyme so the value of luminescence intensity immediately after reaction start and 15 s after reaction start did not vary more than 2%. The measured specific luciferase activity was expressed in relative light units (RLU) s$^{-1}$ M$^{-1}$ of an enzyme. The activity of each enzyme sample was measured in at least three repetitions.

## Measurement of steady-state kinetic parameters of luciferase reaction

Steady-state kinetic parameters of the luciferase reaction of NanoLuc and its mutants were determined at 37 °C using a FLUOStar Omega microplate reader (BMG Labtech, Germany). A series of buffered FMZ (0.05–8.0 μM) and CTZ (0.05–32.0 μM) solutions were prepared by dilution of their ethanolic stock solution into 100 mM potassium phosphate buffer (pH 7.50). Samples of 25 μL of purified enzyme solution were placed in microplate wells. After 10 s baseline collection, the luciferase reaction was initiated by injection of 225 μL of buffered FMZ or CTZ solution and monitored for total luminescence (240–740 nm) for 15 s; this was performed for the entirety of the two luciferin concentration series. The final enzyme concentration was chosen as 0.01 or 0.05 μM depending on the enzyme-luciferin combination, so the enzyme concentration never exceeded 1/5 of the lowest used initial luciferin concentration. To estimate the values of the Michaelis constant ($K_m$) of the four reactions, the obtained dependences of luciferase reaction initial velocity on the luciferin concentration were fitted by nonlinear regression to Michaelis-Menten kinetic model accounting for substrate inhibition using GraphPad Prism 8.4.3 (GraphPad Software, USA). Furthermore, the same measurement was repeated for luciferin concentration levels within the range of 0.25 −4 × $K_m$, only the luminescence of the reaction mixture was monitored either until the luminescence intensity decreased under 0.5% of its maximal measured value (i.e., until the substrate was fully converted to product) or until the reaction has reached the 1000 s time point. In the case that for a certain enzyme-luciferin combination the luminescence never decreased under 0.5% of its maximal

measured value, an additional calibration measurement of luciferin conversion was performed using an excess of enzyme ensuring >99.5% conversion of the added luciferin. Each measurement was performed in at least three repetitions.

Monitoring the luciferase reaction beyond the initial linear phase up to the complete conversion of the substrate allows for the determination of its kinetic constants from reaction rate time progress in relative units without the need for luminometer quantum yield calibration[3]. The measured dependences of luminescence intensity on the reaction time were transformed into cumulative luminescence in time. The obtained conversion curves capturing the initial reaction velocity and total luciferin conversion were globally fitted by numerical methods using the KinTek Global Kinetic Explorer 6.3.170707[49] (KinTek Corporation, USA) to directly obtain the values of turnover number $k_{cat}$, Michaelis constant $K_m$, specificity constant $k_{cat}/K_m$, and equilibrium dissociation constant for enzyme-product complex $K_p$ and enzyme-substrate-substrate complex $K_s$ according to models (Supplementary Fig. 32) for NanoLuc and (Supplementary Fig. 33) for its Y94A mutant. To reflect fluctuation in experimental data, the values of substrate or enzyme concentrations were corrected (±5%) to obtain the best fits. Residuals were normalized by sigma value for each data point. In addition to S.E. values, a more rigorous analysis of the evaluation reliability was performed by confidence contour analysis using FitSpace Explorer[50] implemented in KinTek Global Kinetic Explorer (KinTek Corporation, USA). The scaling factor, relating the luminescence signal to product concentration, was applied as one of the fitted parameters, well defined by the end state of total conversion curves. Depletion of the available substrate after the reaction was verified by repeated injection of a fresh enzyme, resulting in no or negligible luminescence.

### Live-cell imaging assays

ARPE-19 cell line (ATCC; catalog number: CRL-2302) was cultured in Knockout Dulbecco's modified Eagle's medium (Invitrogen, Life Technologies Ltd.) containing 10% fetal bovine serum (FBS), (Biosera), 1× GlutaMAX (Invitrogen, Life Technologies Ltd.), 1× MEM non-essential amino acid solution, 1× penicillin/streptomycin (Biosera) and 10 μM β-mercaptoethanol (Sigma-Aldrich). The cells were incubated at 37 °C/5% CO$_2$ and regularly passaged using trypsin.

Lentiviral particles were generated as described previously[51,52]. Briefly, HEK293T cells were transfected with pSIN vector coding for either NanoLuc$^{CTZ}$ or NanoLuc gene together with second generation of lentiviral production plasmids psPAX2 (Addgene #12260) and pMD2.G (Addgene #12259) kindly provided by Didier Trono. After transfection, the cell culture medium was exchanged for medium containing: OptiMEM (Invitrogen, Life Technologies Ltd.), 1% FBS, 1% MEM non-essential amino acid solution, 1% penicillin/streptomycin) and was collected every 12 h for a total of 48 h. Virus supernatant was centrifuged (4500 × g, 10 min, room temperature), and filtered through a 0.45 μm low protein-binding filter. The supernatant was mixed with Polybrene (Sigma-Aldrich) at a final concentration of 5 μg/mL and applied to cells overnight. The next day, the culture medium containing viral particles was replaced with a fresh medium. Transduced cells were then cultured in the presence of 1 μg/mL puromycin.

For luciferase activity measurements, 250,000 ARPE-19 cells were seeded onto a 40 mm cell culture petri dish, allowed to adhere and recover for 16 h. Cell culture medium was replaced with pre-heated (37 °C) fresh medium containing EnduRen (60 μM) or 1× Nano-Glo Endurazine (Promega). Luciferase activity was measured using LuminoCell device, as described previously[53]. Briefly, luciferase activity was real-time monitored for 24 h using a light-to-frequency converter built in the LuminoCell. Light generated by the luciferase is converted into a series of square-wave pulses, with the frequency depending on the light intensity; thus, the luciferase activity is demonstrated by a number of detected pulses in a given time (integration time).

### RNA isolation and real-time quantitative PCR (RT-qPCR)

Cells were washed with phosphate-buffered saline (PBS), and harvested in 300 μl RNA Blue Reagent (an analog of Trizol) (Top-Bio). RNA was isolated using Direct-zol™ RNA Microprep kit (Zymo Reseach) according to the manufacturer's instructions. RNA was reverse transcribed using a High-Capacity cDNA Reverse Transcription Kit (Applied Biosystems). The RT product was amplified using the Light-Cycler 480 Real-Time PCR system (Roche) using PowerUp SYBR Green Master Mix (Applied Biosystems). Primer sequences used in RT-PCR are shown in Supplementary Table 8. Datasets were normalized to the corresponding levels of GAPDH mRNA.

### Western blotting

Cells were washed three times with PBS (pH 7.4) and lysed in a buffer containing 50 mM Tris-HCl (pH 6.8), 10% glycerol and 1% sodium dodecyl sulfate (SDS). The lysates were then supplemented with 0.01% bromophenol blue and 1% β-mercaptoethanol and denatured at 100 °C for 5 min. The prepared samples were separated by SDS-polyacrylamide gel electrophoresis and transferred onto a polyvinylidene fluoride (PVDF) membrane (Merck Millipore). The PVDF membrane was blocked in 5% skimmed milk in Tris-buffered saline containing Tween for 1 h and incubated with primary antibodies overnight at 4 °C. The following primary antibodies were used: Nano-Luc (N7000, Promega, 1:1000), β-ACTIN (#4970S, Cell Signaling Technology, 1:1000). After an extensive wash in Tris-buffered saline containing Tween the membranes were then incubated with secondary antibodies: Anti-rabbit IgG, HRP-linked Antibody (#7074, Cell Signaling Technology, 1:5000), Anti-mouse IgG, HRP-linked Antibody (#7076, Cell signaling Technology, 1:5000) for 1 h at room temperature. After incubation with ECL (Bio-Rad) the membranes were visualized using ChemiDoc Imaging System (Bio-Rad).

### Preparation of ligand molecules for docking

The structures of the ligands were prepared using Avogadro 1.2.0 software[54]. The multiplicity of the bonds was edited to match the keto forms of FMZ and CTZ, all missing hydrogens were added, and then the structures were minimized by the steepest descend algorithm in the Auto Optimize tool of Avogadro, using the Universal Force Field (UFF). Next, the ligands were uploaded to the RESP ESP charge Derive (R.E.D.) Server Development 2.0[55] to derivate the restrained electrostatic potential (RESP) charges. Then, AutoDock atom types were added and PDBQT files were generated by MGLTools 1.5.4[56,57].

### Preparation of receptor molecules for docking

The NanoLuc structures, which served as receptors for docking (PDB IDs 8AQ6, 8BO9, 5IBO, 5B0U, 7MJB), were downloaded from the RCSB PDB[58], aligned to PDB ID 8AQ6 in PyMOL 1.8.4[59], and stripped of all non-protein atoms. The structures were protonated with H++ web server 4.0[60,61], using pH = 7.4, salinity = 0.1 M, internal dielectric = 10, and external dielectric = 80 as parameters. AutoDock atom types and Gasteiger charges were added to the receptors by MGLTools[56,57] and the corresponding PDBQT files were generated. All receptors were prepared without ligands.

### Molecular docking

The AutoDock Vina 1.1.2[62] algorithm was used for molecular docking. For site-directed docking to the crystallographic binding pocket, the docking grid was selected to be $x = 32$ Å, $y = 22$ Å, $z = 30$ Å sized box with a center in $x = 40$, $y = -47$, $z = 62$ for NanoLuc monomer and a $30 × 24 × 40$ Å box with a center in (35, −50, 60) for NanoLuc dimer covering the catalytic pocket, which was computed with HotSpot Wizard 3.1[63]. For blind docking, a $60 × 50 × 46$ Å box with a center in (47, −57, 63) covering the whole protein was used for the monomer, and a $65 × 75 × 50$ Å box with a center in (44, −44, 62) for the dimer. For site-directed docking to the central cavity inside the β-barrel structure

of NanoLuc, three different cubic boxes with a side of 22, 20, and 18 Å centered in (47, −60, 62) were used as the docking grids. The flag --exhaustiveness = 100 was used to sample the possible conformational space thoroughly. The number of output conformations of the docked ligand was set to 10. The results were analyzed in PyMOL 1.8.4 software[59].

### Ligand preparation for adaptive sampling

The structures of the ligands (FMZ and Cl⁻) were extracted from the NanoLuc crystal structure. The multiplicity of bonds in the FMZ structure was adjusted to match the keto form and all missing hydrogens were added using Avogadro 1.2.0 software[54]. The *antechamber* module of AmberTools16[64] was used to calculate the charges for the ligands, add the atom types of the Amber force field and compile them in a PREPI parameters file. Also, the *parmchk2* tool from AmberTools16 was used to create an additional FRCMOD parameter file for FMZ to compensate for any missing parameters.

### System preparation and equilibration

The structure of the NanoLuc monomer was extracted from the crystallographic structure of the NanoLuc dimer in a complex with FMZ. The crystallographic water molecules were kept in the system. Three starting NanoLuc systems were prepared: (i) monomer + FMZ + two $O_2$ molecules, (ii) dimer + two $O_2$ molecules, and (iii) dimer + FMZ + two $O_2$ molecules.

The following steps were performed with the High Throughput Molecular Dynamics (HTMD) 2[65] scripts. Each protein structure was protonated with PROPKA 2.0 at pH 7.5[66]. For the systems with FMZ, one molecule was placed in the same site as in the initial crystal structure. The three systems were solvated in a cubic water box of TIP3P[67] water molecules with the edges at least 10 Å away from the protein, by the solvate module of HTMD 2. Cl⁻ and Na⁺ ions were added to neutralize the charge of the protein and get a final salt concentration of 0.1 M. The topology of the system was built, using the *amber.build* module of HTMD 2, with the ff14SB[68] Amber force field and the previously compiled PREPI and FRCMOD parameter files for the ligands. The systems were equilibrated using the *equilibration_v2* module of HTMD 2[65]. The system was first minimized using a conjugate-gradient method for 500 steps. Then the system was heated to 310 K and minimized as follows: (i) 500 steps (2 ps) of NVT thermalization with the Berendsen barostat with 1 kcal·mol⁻¹·Å⁻² constraints on all heavy atoms of the protein, (ii) 1,250,000 steps (5 ns) of NPT equilibration with Langevin thermostat and same constraints, and (iii) 1,250,000 steps (5 ns) of NPT equilibration with the Langevin thermostat without any constraints. During the equilibration simulations, holonomic constraints were applied on all hydrogen-heavy atom bond terms, and the mass of the hydrogen atoms was scaled with factor 4, enabling 4 fs time steps[69–72]. The simulations employed periodic boundary conditions, using the particle mesh Ewald method for treatment of interactions beyond 9 Å cut-off, the 1–4 electrostatic interactions were scaled with a factor of 0.8333, and the smoothing and switching of van der Waals interaction was performed for a cut-off of 7.5 Å[71].

### Adaptive sampling

HTMD[65] was used to perform adaptive sampling of the conformations of the three NanoLuc systems (dimer + FMZ + 2 $O_2$, dimer + 2 $O_2$, monomer + FMZ + 2 $O_2$). Fifty ns production MD runs were started with the systems that resulted from the equilibration cycle and employed the same settings as the last step of the equilibration. The trajectories were saved every 0.1 ns. Adaptive sampling was performed using, as the adaptive metric, the root-mean-square deviation (RMSD) of all Cα atoms of the protein against the crystal structure as a reference, and time-lagged independent component analysis (tICA)[73] projection in 1 dimension. 20 epochs of 10 parallel MDs were performed for the two

systems with NanoLuc dimer, corresponding to a cumulative time of 10 μs per system. For the monomer, 29 epochs of 10 MDs were calculated with the same metric (14.5 μs), and additional 4 epochs of 10 MDs (2 μs) were calculated with the contacts metric between all heavy atoms of FMZ and residues 41I, 57H, and 89 K located in the active site of the protein.

### Markov state model construction

The simulations were made into a simulation list using HTMD[65], the water and ions were filtered out, and unsuccessful simulations shorter than 50 ns were omitted. Such filtered trajectories were combined for each system, which resulted in 10 μs of cumulative simulation time for the two systems with NanoLuc dimer and 16.5 μs for the system with NanoLuc monomer. The ligand unbinding dynamics of the systems with FMZ were studied by the RMSD metric for the heavy atoms of FMZ against the initial position of FMZ in the system, and this property was checked for convergence (Supplementary Fig. 34). The data were clustered using the MiniBatchKmeans algorithm to 1000 clusters. For the NanoLuc dimer with FMZ, a 20 ns lag time was used in the models to construct three Markov states, while for the monomer with FMZ three Markov states were constructed using a 30 ns lag time. The dimer dissociation dynamics of the dimer systems were studied by the same metric used in the adaptive sampling—the RMSD of the Cα atoms of the protein, and this property was checked for convergence (Supplementary Fig. 35). The data were clustered using the MiniBatchKmeans algorithm to 1000 clusters. For the NanoLuc dimer with FMZ, a 20 ns lag time was used in the models to construct 3 Markov states, while for the dimer alone, 4 Markov states were constructed using a 20 ns lag time. The Chapman–Kolmogorov test was performed to assess the quality of all the constructed states. The states were visualized in VMD 1.9.3[74] and statistics of the RMSD value for each state were calculated (mean RMSD, SD, minimum RMSD, and maximum RMSD). The trajectory was saved for each model.

### Calculation of kinetics

Kinetic values (MFPT on/off, $k_{on}$, $k_{off}$, $\Delta G^0_{eq}$, and $K_D$) were calculated by the *kinetics* module of HTMD[65] between the source state and the sink state. In the FMZ unbinding analysis, the source state was defined as the unbound state of FMZ and the sink state as the bound state, while in the dimer dissociation analysis, the source state was defined as the most dissociated state and sink as the associated state. Also, the equilibrium population of each macrostate was calculated and visualized. Finally, bootstrapping of the kinetics analysis was performed, using randomly selected 80% of the data, run 100 times. The kinetic values were then averaged, and the standard deviations were calculated.

### Preparation and minimization of NanoLuc structures for ASMD

Six different NanoLuc crystal structures were studied: (i) an asymmetric dimer based on PDB ID 8AQ6 with bound FMZ, (ii) a symmetric dimer of NanoLuc-Y94A mutant (PDB ID 8AQH), (iii) a symmetric dimer of NanoLuc-R164Q mutant (PDB ID 7MJB), and three monomeric structures: chains A of PDB IDs (iv) 8AQ6, (v) 5B0U, and (vi) 8BO9. First, the structures were stripped of all non-protein atoms. Next, the structures were protonated with the H++ web server v. 4.0[60,61], using pH = 7.4, salinity = 0.1 M, internal dielectric = 10, and external dielectric = 80 as parameters. Then, the crystallographic water molecules were added to the systems. Next, histidine residues were renamed according to their protonation state (HID – Nδ protonated, HIE – Nε protonated, HIP – both Nδ and Nε protonated). The FMZ and CTZ ligands were prepared as described in Ligand preparation for adaptive sampling above. Moreover, it was minimized by the steepest descent algorithm in the Auto Optimize tool of Avogadro 1.2.0[54], using the Universal Force Field (UFF). In the case of (i), two systems were prepared—one with FMZ and one without, while systems (ii) and (iii) were

prepared without FMZ. NanoLuc monomers (iv)–(vi) were prepared with CTZ in two different starting positions (at the mouth of the two hypothetical access tunnels). The *tLEaP* module of AmberTools16[75] was used to neutralize the systems with Cl⁻ and Na⁺ ions, import the ff14SB force field[68] to describe the protein and the ligand parameters, add an octahedral box of TIP3P water molecules[76] to the distance of 20 Å from any atom in the dimeric systems and 10 Å in the monomeric systems, and generate the topology file, coordinate file, and PDB file. Crystallographic water molecules overlapping with the protein or ligand were removed from the input PDB file and *tLEaP* was rerun.

The system equilibration was carried out with the *PMEMD.CUDA*[77–79] module of Amber 16[75]. In total, five minimization steps and twelve steps of equilibration dynamics were performed. The first four minimization steps were composed of 2500 cycles of the steepest descent algorithm followed by 7500 cycles of the conjugate-gradient algorithm each, while gradually decreasing harmonic restraints. The restraints were applied as follows: 500 kcal·mol⁻¹·Å⁻² on all heavy atoms of the protein and ligand, and then 500, 125, and 25 kcal·mol⁻¹·Å⁻² on protein backbone atoms and ligand heavy atoms. The fifth step was composed of 5000 cycles of the steepest descent algorithm followed by 15,000 cycles of the conjugate-gradient algorithm without any restraint.

The equilibration MD simulations consisted of 12 steps: (i) first step involved 20 ps of gradual heating from 0 to 310 K at constant volume using Langevin dynamics, with harmonic restraints of 200 kcal·mol⁻¹·Å⁻² on all heavy atoms of the protein and ligand, (ii) ten steps of 400 ps equilibration Langevin dynamics each at a constant temperature of 310 K and a constant pressure of 1 bar with decreasing harmonic restraints of 150, 100, 75, 50, 25, 15, 10, 5, 1, and 0.5 kcal·mol⁻¹·Å⁻² on heavy atoms of protein backbone and ligand, and (iii) the last step involving 400 ps of equilibration dynamics at a constant temperature of 310 K and a constant pressure of 1 bar with no restraint. The simulations employed periodic boundary conditions based on the particle mesh Ewald method[80] for treatment of the long-range interactions beyond the 10 Å cut-off, the SHAKE algorithm[81] to constrain the bonds that involve hydrogen atoms, and the Langevin temperature equilibration using a collision frequency of 1 ps⁻¹. After the equilibration, the number of Cl⁻ and Na⁺ ions needed to reach 0.1 M salinity was calculated using the average volume of the system in the last equilibration step. The whole process was repeated, from the *tLEaP* step, to correct the number of the added ions.

### Adaptive steered molecular dynamics

The dimer dissociation and ligand binding trajectories were calculated with adaptive steered molecular dynamics (ASMD). The ASMD method applies constant external force on two atoms in the simulated systems. This can be used to either push two atoms from each other or pull them together to simulate unbinding/binding of ligands or dissociation/association of proteins. During ASMD, several parallel simulations are started from the same state. The simulation runs in stages where a chosen value changes the distance between selected atoms. At the end of each stage, the parallel simulations are collected and analyzed, and the Jarzynski average is calculated. The trajectory with its work value closest to the average is selected and the state at the end of this trajectory is used as the starting point for the next stage. For our purpose, we used the default values for setting up ASMD which were found in the tutorial for AMBER and the ASMD publication[82]. The simulations were run with 25 parallel MDs, steered by 2 Å stages of distance increments, with a velocity of 10 Å/ns, and a force of 7.2 N. The rest of the MD settings were set as in the last equilibration step. For the dimer dissociation, the atoms selected for steering were Cα atoms of I58 residues from the dimer interface so that the two subunits could be pushed apart. This residue is part of a β-barrel structure, which makes it suitable for steering since the structure is relatively rigid. The distance between the two Cα atoms was measured in the last snapshot

from the equilibration MD using Measurement Wizard in PyMOL 1.8.4[59] and the two subunits were steered apart for an additional 20 Å.

For the ligand binding in the central cavity of NanoLuc, the steering atoms were Cα of G116/ O28 of CTZ when using the first tunnel (between H2/H3), Cα of W161/C19 of CTZ for the second tunnel (between H3/H4). The starting distance of the steering atoms was measured in the equilibrated systems, and the ligand was pulled inside the protein to the same distance as in the NanoLuc^CTZ crystal, which served as a reference. The ligand binding simulations were run in triplicate and the lowest-energy replica was selected as representative. MD trajectories were analyzed and visualized in PyMOL 1.8.4[59] using the smooth function and exported as movie (.mpg) files.

### Reporting summary

Further information on research design is available in the Nature Portfolio Reporting Summary linked to this article.

## Data availability

Atomic coordinates and structural factors have been saved in the Protein Data Bank (www.wwpdb.org)[20] under PDB ID accession codes: 8AQH, 8AQI, 8AQ6, and 8BO9. Previously published structural coordinates used in this study 7MJB, 7VSX, 5B0U, and 7SNT were downloaded from the Protein Data Bank (www.wwpdb.org)[20]. SAXS datasets, experimental details, atomic model, and fits have been saved in the Small-Angle Scattering Biological Data Bank (www.sasbdb.org) as entry SASDSQ9. The primary data from molecular dynamics simulations are available in Zenodo repository with the identifier https://doi.org/10.5281/zenodo.8302143. Source Data are provided with this paper.

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

## Acknowledgements

The work on this paper was supported by the Czech Science Foundation (GA22-09853S) and the Czech Ministry of Education (INBIO CZ.02.1.01/0.0/0.0/16_026/0008451, RECETOX RI LM2023069, e-INFRA LM2018140, CETOCOEN EXCELLENCE CZ.02.1.01/0.0/0.0/17_043/0009632). This project has received funding from the European Union's Horizon 2020 research and innovation program under grant agreement No 857560 (CETOCOEN Excellence). Supported by the European Union (No. 101087124, ADDIT-CE). This publication reflects only the author's view, and the European Commission is not responsible for any use that may be made of the information it contains. M.T. is a Brno PhD Talent Scholarship holder funded by the Brno City Municipality. CIISB research infrastructure project (LM2018127) is acknowledged for financial support of the measurements at Biomolecular Interactions and Crystallization Core Facility. G.G. acknowledges a PhD fellowship from the Université Paris Descartes, Sorbonne Paris Cité, France and we also benefited from the Valoexpress funding calls of the Institut Pasteur, Paris. The crystallographic data were collected at the PXIII beamline at the Swiss Light Source (SLS) in Villigen (Switzerland). We are grateful to the members of the SLS synchrotron for the use of their beamline and help during data collection. We thank Andrea Smith (MU) for help during the X-ray data collection, and Tomas Klumpler (CEITEC-MU, Brno, Czech Republic) for assistance during SAXS data collection and processing.

## Author contributions

M.N., T.S., J.T., V.N. and K.S. prepared and cloned the DNA constructs, and produced the protein samples for crystallization and biochemical experiments. R.B., G.G., Y.L.J. designed and synthesized the luciferin analogs. M.N. carried out the crystallization screenings, and optimized crystallization hits. M.N. and M.Mar. collected diffraction data and solved the protein crystal structures. D.P., T.S., M.Maj., Z.P. performed luminescence measurements and kinetic analyses. T.B. engineered stable mammalian cell lines and performed cell-based bioluminescence experiments. J.H., S.M.M., J.D. and D.B. carried out computational and theoretical experiments and MD simulations. M.Mar., Y.L.J., Z.P., J.D., M.T., D.B. and Z.P. designed the project, supervised research and interpreted data. M.N. and M.Mar. wrote the manuscript with contributions from all authors. All authors have approved the final version of the manuscript.

## Competing interests

The authors declare no competing interests.
