## [Peer Review File · Nature Communications]

REVIEWER COMMENTS

Reviewer #1 (Remarks to the Author):

This paper reports on a comprehensive study on the NanoLuc system after finding ligands that binds well toward its crystallization. In general the contents make sense and the explanation is plausible. Simply, the structures tell almost everything about what the authors are describing. Readers who are interested in the NanoLuc system will definitely be interested in this work.

A few minor complaints that I have are:

1. The manuscript is rather long. To me it was to an uncomfortable or even painful extent. I believe some can be shifted to SI.
2. Dimerization and tetramerization are explained to be crucial for allosteric aspects of the enzyme system. Can the authors suggest some FRET type of experiments that can further prove what they are proposing? This may be important in proving that the the mechanism of the multimer formation is not an artifact of crystallization.
3. The experiments suggest a communication pathway between allosteric and catalytic sites. More discussion will be helpful for readers. Yes, structural data support it, but that is not really the biological story behind it. And readers like me would like the biological story a lot better.

Reviewer #2 (Remarks to the Author):

The authors determined the crystal structures of nanoLuc luciferase in complex with reaction products CEI or FMA, and found the ligands bound to the surface pocket of the enzyme other than the expected catalytic center inside the protein. It was anticipated via comparisons with apo-form structures that the binding of CEI/FMA had allosteric effect for the enzyme by altering the conformation of several catalytic residues, and the bindings of a substrate to catalytic center and a product to allosteric site are mutually exclusive, suggesting a homotropic negative allostery mechanism. The authors also engineered the allosteric site and succeeded in improving nonoLuc bioluminescence with CTZ. They also determined the complex structure of nonoLuc and substrate analogue azaCTZ to predict the reaction mechanism of the enzyme, and executed the docking and molecular dynamics simulations to predict the entrance pathway

of the substrate to the catalytic center. The experiments were, basically, done carefully and appropriately, and a considerable amount of interesting solid data were provided as the results. This reviewer, however, concerned the insufficiency of the current manuscript in general and interdisciplinary perspective, which should be required to attract wide variety of researchers/readers of this journal, as detailed in the attached comments. The authors should thoroughly improve their presentation on this point, which would be mandatory for the presented study to be recommended for publication in this journal.

Major points:

1) Lines 568-571: "we hypothesize that such inactive enzyme-luciferin complexes could function as a storage pool of luminescent agents ...", and lines 579-580: "It was unclear whether the binding of luciferin molecules to the surface ..." Although a lot of solid results were presented, the findings such as homotropic negative allostery were within the already-existing concepts, and almost nothing was discussed about how the findings contribute to a new concept in the biology/molecular biology, which would be mandatory to attract a wide-variety of readerships of this journal. For example, a schematic diagram of entire picture including allosteric regulation and catalytic reaction that explain how the observed allostery works biologically/enzymologically in the system should be provided and fully discussed. Otherwise, this manuscript would be rather suitable for other specialized structural/molecular biology journals.

2) Lines 563-566: "This could explain the fact that NanoLuc is monomeric enzyme ... may trigger its homodimerization/homotetramerization ..." Any experimental clues, for example solution scattering or size-exclusion chromatography, were not provided in the presented manuscript. This point largely prevents the findings from being positioned in a bigger picture. This point should be appropriately resolved.

3) Lines 688-689: "Concentrated NanoLuc was mixed with a 4 molar excess of furimazine non-oxidizable analogue azaFMZ " This reviewer could not understand the meaning of this procedure. If the azaFMZ molecule was not observed neither in allosteric site nor catalytic center, why this molecule did not bind to the protein? What will happen if this additive was not provided? Lines 695-696: "The good-looking crystals were soaked overnight in the mother liquor supplemented with 10 mM FMZ or CTZ." It seemed the observed ligands in the crystal structures were added as substrates and reactions took place in the crystallization solutions, and finally the reaction products bound to the allosteric site. If this is correct, this reviewer also did not understand the necessity of this procedure. Why not add the products (CEI or FMA) from the beginning? These points would be very sensitive in interpreting the observed crystal structures, and fully and explicitly explained in the main text.

Minor points:

1) Line 150: "the R1 2-(furan-2-yl) and R3 8-benzyl", line 180: "tail part", line 202: "2-(furan-2-ylmethyl)", or line 203: "8-benzyl", for example. The references to the molecular moieties in this manuscript would be difficult for many of the readers to follow. The molecular formulas for the major ligands, for example CEI, FMA, CTZ, or FMZ, on which mentioned moieties were indicated, should be provided.

- 2) Lines 161-162 : "several pocket-shaping residues (e.g. E4, R43 and 162 R166), played a key role during the development of NanoLuc." The pocket-shaping residues should be indicated in Fig. 1i.
- 3) "Scheme 1" in Suppl Fig. 6 legend was sudden and would get lost in context.
- 4) "CBS-1 and CBS-2" in Fig. 4 legend would be the abbreviation of chloride binding site but not defined anywhere.
- 5) "contour level 1.0 σ " in Fig. 6a legend. Generally, this threshold is too low. Also the density is not well defined and not convincing that a model based on this density could be used to discuss the reaction coordinates. This figure should be revised with higher threshold ($> 3.0 \sigma$) and the map for entire system including protein must be shown without trimming.
- 6) Fo-Fc maps at catalytic central cavity for the crystal structures of NanoLuc/CEI, NanoLuc/FMA, and NanoLuc-Y94A should be presented to confirm the emptiness of the central cavity.
- 7) The descriptions of reaction in lines 390-406 and Fig. 6e legend are difficult to follow. The schemes in Fig. 6e should be labeled with serial numbers (e.g., 1, 2, 3[~], or i, ii, iii[~]) and the numbers should be referred to in the texts.
- 8) The first and second tunnels should be indicated on the structure models in Fig. 7a or 7e.
- 9) Molecular formulas in lines 626 and 637 were not clear from where these formulas were referred to. The reference points should be explicitly made in the text.
- 10) Line 693: "azafurimazine" or lines 745-746 "furimazine and coelenterazine" for example. These molecules would be better presented in the abbreviations.

Reviewer #3 (Remarks to the Author):

The authors investigate the widely used engineered luciferase NanoLuc and describe allosteric relations using new structural insights obtained from co-crystal structures with the reaction products of CTZ and FMZ oxidation, and the unreactive substrate analogue aza-coelenterazine. The reaction products are found to bind at an allosteric binding pocket formed at the interface of two NanoLuc domains. In contrast, aza-coelenterazine is bound in the proposed intramolecular, catalytically active binding site. Based on these and previously derived structures, modeling and site-directed mutagenesis, the authors derive and propose a detailed mechanism for substrate conversion of NanoLuc and its allosteric regulation. In addition, novel NanoLuc variants are obtained with enhanced activity with the classical substrate coelenterazine that rival the activity of NanoLuc with its currently used substrate furimazine. The manuscript is well written and structured and provides key insights for the future improvement of NanoLuc and its applications. While it remains to be established whether the allosteric pocket formed at the dimer-interface observed in the X-ray structure is also formed in solution (see also below), the work is clearly important in a sense that it provides evidence for substrate/product controlled switching between a catalytically-active open and a catalytically-inhibited closed active site. The work is therefore

recommended for publication in Nature Communications after minor revisions. Comments are listed below.

Major questions and comments

1. A key remaining question is whether the NanoLuc dimer observed in X-ray structures is also formed in solution, in particular at the sub/low nM concentrations that NanoLuc is normally used/studied. The modeling studies reported here suggests that substrate/product could act as a molecular glue to promote dimerization, but no experimental evidence is provided (or cited from the literature). Can the authors comment on how to address this question experimentally?

2. In the absence of experimental verification, the observed binding pockets could still be the results of the very high concentration of NanoLuc monomers in the crystal/ crystallization artefact. The results of the various mutagenesis studies provide more insight into the importance of several key residues and even yield some very useful improved variants, but by themselves do not unequivocally proof the existence of the dimer in solution. The observed conformational changes may still be relevant and hint at a regulatory role in the monomer in solution. It may be good for the authors to discuss the possibility that products are also inhibiting or preferably binding the surface pocket to allosterically regulate the open/close conformational change. Are there any reports in literature that describe product inhibition of NanoLuc?

3. It would be helpful if the authors could discuss the implications of their findings for biosensor design and other applications of NanoLuc. E.g. the authors write that the extension of NanoLuc C-terminus by two alanines 'substantially decreased' CTZ luminescence. Does this imply that C-terminal fusion proteins of this variant, or the new NanoLucCTZ variant, may show reduced luciferase activity when using CTZ? Is this consistent with literature (NanoLuc fusion proteins?) or has this not been described before? What are the implications of their finding for applications involving the split luciferase NanoBiT system?

Minor/technical questions/remarks:

- Line 112/113 renewable light-producing technologies. Why would this help engineering renewable light-producing technologies? How would these look like?

- Line 171 Figure 1 F: The chord plot is simply mentioned but the main message is not explained in the caption. A short additional explanation in one sentence would be helpful, as it is also very condensed in the main text

- Figure 3b. No added value in showing the numbers of k_{cat} and K_m that are already presented in Table 1 also as (3 different!) graphs.

- Line 242 'mechanisms of communication' sounds a bit broad, potentially a better phrasing would be 'mechanism of allosteric importance'?

- Line 259/260 Why is the flipping of beta strand S5 called 'unusual'? Because it requires an extreme conformational change? Are there any comparable cases that show a beta strand flipping?

- Line 264 change to enzyme-substrate Michaelis complex, as Michaelis complex is a connected term / definition. –
- Line 281 figure caption (iii) for FMA-bound NanoLuc structure
- Line 324 Might be useful to explain the reason of using EnduRen instead of original coelenterazine? Is EnduRen still cheaper and 'more available' than Endurazine, the used derivate of Furimazine?
- For in vitro assays, it would be useful to provide absolute luminescent intensities of both NanoLuc and NanoLucCTZ at the same concentrations and gain, each with both substrates, in vitro to allow a direct comparison of their behaviour. Alternatively, raw data of intensities like photon counts/s should be provided in Fig 5 D, E (line 297)
- Fig 5 D/E: What determines the time-dependence of these signals. It is not clear if and how one compare the values on the Y-axes between d and e, in particular as they are depicted as a.u. However, the scaling between 0 and 100 suggests that some kind of normalization was performed, also because the signal with FMZ is reported to be 50%. The conclusion that the engineered NanoLucCTZ and EnduRen substrate represent a superior luciferase-luciferin reporting pair for long-term live-cell imaging applications require more extensive experiments, e.g, showing similar expression levels for both variants.
- Unfinished sentence in line 464/465
- Line 521, write out fatty acid binding protein at first mention of FABP, as done later for nsLTPs.
- Line 528 mention of initial crystallization experiments. The authors did not provide any data regarding this statement, these could be added to the supporting information.
- Line 564 NanoLuc is a monomeric...
- Discussion line 568/569: Interesting hypothesis, but how would the luminescent agents storage be generated in the first place if at lower concentrations the substrate is converted? In additional presence of very high chloride concentration/halide ions? Although this is mentioned later in the paragraph, the connection between these statements is not very clear.
- Line 579 surface allosteric site could be a ...
- Supplement, line 320 reference not listed
- (Methods) Line 673 Was the expression from 10 ml of 2xLB enough to provide high yields for the NanoLuc variants, or is the large culture not mentioned?
- (Methods) Line 684 Please add at least one SDS-PAGE to the supplement to confirm purity of the NanoLuc variants.
- (Methods) Line 740 Are the mentioned relative light units (RLU) in s⁻¹ M⁻¹ related to the A.U. shown in Fig. 5 (line 297)?

A point-by-point response to comments raised by all reviewers

Reviewer #1:

This paper reports on a comprehensive study on the NanoLuc system after finding ligands that binds well toward its crystallization. In general, the contents make sense and the explanation is plausible. Simply, the structures tell almost everything about what the authors are describing. Readers who are interested in the NanoLuc system will definitely be interested in this work.

Minor:

1. The manuscript is rather long. To me it was to an uncomfortable or even painful extent. I believe some can be shifted to SI.

Answer: Thanks for this comment. We agree with the reviewer that some parts of the manuscript were too long, making it uncomfortable to read. To address this point, we made a re-arrangement of the manuscript, in order to make it more readable. An addition, some less important parts of the results were transferred to the SI file, to make the main-text shorter.

However, the NanoLuc luciferase is the small protein, but the biology behind it is very complex and unusual, and therefore, it requires full explanations and comprehensive descriptions, not to miss important aspects.

2. Dimerization and tetramerization are explained to be crucial for allosteric aspects of the enzyme system. Can the authors suggest some FRET type of experiments that can further prove what they are proposing? This may be important in proving that the the mechanism of the multimer formation is not an artifact of crystallization.

Answer: We thank for this highly relevant fundamental comment. The NanoLuc luciferase (16-point mutant of a catalytic unit of *Oplophorus gracilirostris* luciferase) was engineered as a monomeric enzyme (Hall et al. 2012). In general, all crystallographic structures of NanoLuc luciferase determined so far can be divided into two groups. The first group represents so-called “closed-barrel” NanoLuc structure, where intra-barrel (catalytic cavity) is strongly reduced, preventing luciferin binding to the catalytic site. Instead, there is a ligand-binding pocket shaped on the enzyme surface, where diverse ligand molecules can bind (fatty acid, polyethylene glycols, or luciferins). In our crystallographic experiments, we were able to capture either furimamide or coelenteramide bound in this surface allosteric site. The structures crystallized in the “closed-barrel” NanoLuc form are characterized by tight homotetrameric association, where the bound ligands are found at the protein-protein interface. On the other hand, the second group of NanoLuc structures represents so-called “open-barrel” NanoLuc structure, where the surface allosteric site is disappeared, and instead, a voluminous ligand-binding pocket is formed inside the β -barrel structure. Based on our azacoelenterazine-bound NanoLuc structure, we think that this intra-barrel pocket is a real catalytic site where luciferase reaction occurs. This assumption is now also supported by

recently deposited 3-methoxy-furimazine- and inhibitor-bound NanoLuc X-ray structures (PDB IDs: 7SNT and 7SNW; *unpublished data*). In the manuscript revision, we performed additional structure-based mutagenesis in the intra-barrel catalytic site to validate the reaction mechanism that we are proposing in the manuscript. All mutagenesis data are in agreement with structural observation and biochemical conclusions.

Obviously, the NanoLuc luciferase is capable to undergo a large conformational change, between the “closed-barrel” and “open-barrel” structures. We show that a binding to the allosteric site prevents simultaneous binding to the catalytic site, and vice versa. We experimentally evidence that restructuring of the allosteric site can boost the luminescent reaction in the remote intra-barrel active site, demonstrating that the surface allosteric site is vital for enzyme function.

The fact that all crystallographic structures of NanoLuc luciferase in its “closed-barrel” structure show tight homotetrameric association is puzzling. In the manuscript, we do not claim that the dimerization/tetramerization is crucial for allosteric aspects of the enzyme system. We aimed to describe crystallographic structures thoroughly, in a broader picture, because this system is very unusual. Indeed, it can be a crystallographic artefact, as highlighted by the reviewer. We agree with this concern, and we do not definitively want to over-interpret the observation. What we clearly see is a remarkable conformational transition between the “closed-barrel” and “open-barrel” NanoLuc structure. When adopting the “closed-barrel” conformation, a ligand-binding pocket forms on the enzyme surface. Previous studies showed that this pocket can accommodate a fatty acid molecule. In our experiments, we found bound luciferin molecules, but also PEG molecule, originating from the crystallization buffer. To what extent the crystal packing forces and/or other factors are responsible for the tight homotetrameric association seen in all NanoLuc crystals with the “closed-barrel” is not clear. Therefore, we always use terms like “crystallographic dimer” or “crystallographic tetramer” in our text. These “crystallographic oligomers” can indeed be an artificial consequence of crystal packing that would not be maintained in the solution phase. A puzzling question is then why do single point mutations targeting the surface ligand-binding site selectively boost the bioluminescence with native CTZ luciferin, but not with the synthetic one? Our experimental results point to that the surface ligand-binding site is vital for enzyme function.

To further dissect this issue, during the revision of this manuscript, we performed additional small-angle X-ray scattering (SAXS) experiments with NanoLuc luciferase in absence and presence (4 molar excess) of the luciferin molecule to probe oligomeric state of NanoLuc luciferase in action. The SAXS experiment showed that the NanoLuc luciferase exists as a monomeric enzyme in standard micromolar concentration, both with and without luciferin molecules. The SAXS results were added into the manuscript. In-solution experiments, in a condition when the NanoLuc luciferase is used in diverse bioassays, showed that the enzyme is indeed monomeric. We thus think that the conformational transition is important for enzyme catalysis and/or its regulation. For instance, the large conformational change can be essential for un/loading of relatively large and hydrophobic luciferin from the intra-barrel catalytic site.

Moreover, we **ONLY** hypothesize that the conformational transition, coupled with an oligomeric change (monomer-to-tetramer transition captured in several crystal structures), could be a mechanism for enzyme catalysis inactivation. We anticipate that this scenario could happen in vivo, when native *Oplophorus* luciferase is stored in large concentrations in glands at the base of antennae. We can imagine that in such “extreme conditions” this could be the case here, to store luciferase/luciferin complexes in inactive states (“closed-barrel”). However, this is just the hypothesis. In our experiments, we worked with engineered NanoLuc luciferase (16 mutations introduced, Hall et al. 2012), but not with native *Oplophorus* luciferase that indeed shows complex quaternary structure. Future work with native *Oplophorus* luciferase can only explore this hypothetical model. However, the native *Oplophorus* luciferase is poorly soluble, unstable, displaying complex quaternary structure, making it a difficult target for routine biochemistry studies.

Major findings of this manuscript are: (i) localization of NanoLuc luciferase catalytic site and allosteric site through comprehensive crystallographic experiments, (ii) mapping the molecular recognition between enzyme and luciferin molecules, (iii) delineation of NanoLuc catalytic reaction mechanism, and (iv) highlighting its conformational transition, where we yielded improved luciferase variant (NanoLuc^{CTZ}) via an engineering of the allosteric site on the enzyme surface. Additionally, we provide a hypothetical model for the active-to-inactive switch, to stimulate future directions in the research. The manuscript was modified accordingly, toned down with some conclusions on oligomerization. Our SAXS experiments showed that the NanoLuc luciferase exists in a solution (micromolar concentrations) as a monomeric protein.

We added our structure-based hypothesis as a new part to the Results section, in order to synthesize all our results:

A structure-based model for NanoLuc luciferase action

By using X-ray crystallography, we could visualize in this work some luciferin-bound states of NanoLuc luciferase, and thus secure additional insights into the puzzle of its molecular bioluminescence mechanism. From these, along with previously captured crystallographic snapshots as well as available biochemical and computational data, we are providing a structure-based model for this luciferase catalysis (**Fig. 7**). The most striking feature of this model is based on a conformational transition between the so-called “open β -barrel” and “closed β -barrel” structures. Our crystallographic data reveal that the luciferin substrate can bind to an intra-barrel catalytic site (“open β -barrel”) as well as to a secondary allosteric site situated on the surface of the “closed β -barrel” conformation. Moreover, luciferin binding to the allosteric site will prevent its binding to the catalytic site, and *vice versa*, thanks to this conformational transition.

Our model assumes that when the substrate (luciferin) enters the intra-barrel catalytic site (**Fig. 7i**), it is catalytically converted into the product (oxyluciferin) and this is followed by

the emission of a blue photon (**Fig. 7ii**). The structure-based mutagenesis experiments described in this work actually confirmed this assumption. Following this, the release of the reaction product from the intra-barrel catalytic site may be facilitated by an open-to-closed conformational transition (**Fig. 7iii**). At this stage, the reaction product may actually shift to the allosteric site appearing on the protein surface upon this conformational change. Our structural and kinetic data indicate that the native oxyluciferin (CEI) uses a radically different binding mode, involving more extensive interactions, with the allosteric site than the synthetic one (FMA). This could explain why our structure-based restructuring of the allosteric site substantially boosted the CTZ-based luminescence, but had no effect on the FMZ-based luminescence. Our kinetic measurements showed that NanoLuc efficiently binds FMZ ($K_m = 0.123 \pm 0.004 \mu\text{M}$) with a minimal product inhibition ($K_p = 0.56 \pm 0.01 \mu\text{M}$), whereas a less effective binding ($K_m = 0.57 \pm 0.02 \mu\text{M}$) and substantial product inhibition ($K_p = 0.256 \pm 0.005 \mu\text{M}$) is observed with native CTZ-luciferin. We suggest that the oxyluciferin product binding to the surface luciferin-binding site provides an allosteric-based negative feedback loop, restraining the β -barrel opening and halting the reaction after several cycles. This would lead to a flash-type bioluminescence, which, unlike bacterial and fungal bioluminescence, is rather typical of marine one. Therefore, we think that the combination of the engineered NanoLuc luciferase with the synthetic FMZ-luciferin reported by Hall and co-workers⁸ actually removed this allosteric-based negative feedback leading to a so-far unmatched glow-type bioluminescence. Our model also assumes that upon the ligand release from the allosteric site, the free enzyme will regenerate via a closed-to-open β -barrel transition (**Fig. 7iv**).

Finally, NanoLuc luciferase was engineered as an FMZ-using monomeric enzyme⁸, while the corresponding native CTZ-using full-length counterpart, i.e., the catalytic subunit of *Oplophorus* luciferase, displays complex quaternary structure⁷. Here, we have demonstrated with SAXS-based experiments that indeed the NanoLuc exists as a monomeric protein in the micromolar concentration range (**Supplementary Fig. 6**), typical of the conditions of its use in laboratory^{8,12}. Nevertheless, an intriguing feature is that in all the crystallographic structures of NanoLuc, the “closed β -barrel” conformation adopted points out a tightly packed homotetrameric association. This could be a crystallization artifact due to the high concentration of enzyme monomers present in the crystallization drop. However, we hypothesize that this tetramerization may reflect the inherent feature of native *Oplophorus* luciferase although it would have been mostly suppressed in the engineered NanoLuc. In fact, out of the 16 mutations introduced in NanoLuc, up to 7 mutations are localized on the protein surface area involved in the protein-protein contacts seen in the crystallographic tetrameric arrangement. As experimental evidence to support this hypothesis, we constructed and analyzed several NanoLuc “reverse” mutants. As seen by gel filtration, it turned out that some single-point mutants, such as R11E and R43A, did not exist as a pure monomeric protein anymore but rather as monomer-tetramer mixtures (**Supplementary Fig. 10**). We also speculate that in “real life”, such tetrameric complexes could serve as an “inactive” luciferin-loaded storage form of *Oplophorus* luciferase (**Fig. 7v**).

Fig. 7. Structure-based model for NanoLuc luciferase action. I. A substrate molecule enters the intra-barrel catalytic site, the luciferase retains an “open β -barrel” conformation. II. Catalytic conversion of the substrate into its reaction product, followed by the emission of a blue photon. III. Release of the product out of the intra-barrel catalytic site. This step is accompanied by an open-to-closed conformational transition, and a rebinding of the product to the newly formed allosteric site on the protein surface. IV. Dissociation of the product from the surface allosteric site, allowing recycling into the pre-catalytic “open β -barrel” state. V. Alternatively, the “closed β -barrel” structure of NanoLuc luciferase is permissive to undergo a monomer-to-tetramer transition in crystallization conditions. This step is hypothetical and could only be valid for native *Oplophorus* luciferase which exhibits complex quaternary structure. The PDB ID codes for representative crystallographic structures are provided.

3. The experiments suggest a communication pathway between allosteric and catalytic sites. More discussion will be helpful for readers. Yes, structural data support it, but that is not really the biological story behind it. And readers like me would like the biological story a lot better.

Answer: Thanks for this comment and your interest in the story. We must highlight again that we worked with the NanoLuc luciferase engineered (16 mutations introduced) towards furimazine (FMZ) luminescence, but not with native *Oplophorus* luciferase utilizing coelenterazine (CTZ). Therefore, we have to be very careful with any biological conclusion relevant for in vivo situation, although we love biological stories too.

What our experiments clearly showed is the fact that a single-point mutation in the allosteric surface site can substantially, as well as selectively, increase bioluminescence with native coelenterazine, but not with its synthetic counterpart (furimazine). This is very impressive and unexpected observation, and it somehow verifies biological relevance of our crystallographic structures in which the native luciferin binds to the surface allosteric site in a

radically different binding mode than the synthetic one (furimamide). The three most effective mutations (D9R, H57A and K89R) were generated to interfere with the luciferin binding to the allosteric site. The results thus suggest that the ligand-binding pocket on the enzyme surface (allosteric site) evolved to accommodate native luciferin, but not the synthetic one. This can explain why the restructuring mutations selectively boosted bioluminescence with CTZ, but not with the FMZ. And *vice versa*, these experiments provide a new clue explaining why the synthetic FMZ luciferin is so powerful luciferin when combined with the NanoLuc luciferase, which was never highlighted before. The construction of NanoLuc luciferase variants via a restructuring of the allosteric site is one of the novelties of our work. We outline that future engineering of the allosteric site may yield a next-generation of luciferase/luciferin pairs outperforming the current systems. Specifically, our results provide key insights that should aid in the design of mechanistically differentiated ligands (luciferins). The corresponding text of manuscript was accordingly modified.

Finally, our kinetic measurements showed that NanoLuc efficiently binds FMZ ($K_m = 0.123 \pm 0.004 \mu\text{M}$) with a minimal product inhibition ($K_p = 0.56 \pm 0.01 \mu\text{M}$), whereas a less effective binding ($K_m = 0.57 \pm 0.02 \mu\text{M}$) and substantial product inhibition ($K_p = 0.256 \pm 0.005 \mu\text{M}$) is observed with native CTZ-luciferin. We suggest that the oxyluciferin product binding to the surface luciferin-binding site provides an allosteric-based negative feedback loop, restraining the β -barrel opening and halting the reaction after several cycles. This would lead to a flash-type bioluminescence, which, unlike bacterial and fungal bioluminescence, is rather typical of marine one.

Reviewer #2:

The authors determined the crystal structures of NanoLuc luciferase in complex with reaction products CEI or FMA, and found the ligands bound to the surface pocket of the enzyme other than the expected catalytic center inside the protein. It was anticipated via comparisons with apo-form structures that the binding of CEI/FMA had allosteric effect for the enzyme by altering the conformation of several catalytic residues, and the bindings of a substrate to catalytic center and a product to allosteric site are mutually exclusive, suggesting a homotropic negative allostery mechanism. The authors also engineered the allosteric site and succeeded in improving nonoLuc bioluminescence with CTZ. They also determined the complex structure of nonoLuc and substrate analogue azaCTZ to predict the reaction mechanism of the enzyme, and executed the docking and molecular dynamics simulations to predict the entrance pathway of the substrate to the catalytic center. The experiments were, basically, done carefully and appropriately, and a considerable amount of interesting solid data were provided as the results. This reviewer, however, concerned the insufficiency of the current manuscript in general and interdisciplinary perspective, which should be required to attract wide variety of researchers/readers of this journal, as detailed in the attached comments. The authors should thoroughly improve their presentation on this point, which

would be mandatory for the presented study to be recommended for publication in this journal.

Major:

1. Lines 568-571: "we hypothesize that such inactive enzyme-luciferin complexes could function as a storage pool of luminescent agents ...", and lines 579-580: "It was unclear whether the binding of luciferin molecules to the surface ..." Although a lot of solid results were presented, the findings such as homotropic negative allostery were within the already-existing concepts, and almost nothing was discussed about how the findings contribute to a new concept in the biology/molecular biology, which would be mandatory to attract a wide-variety of readerships of this journal. For example, a schematic diagram of entire picture including allosteric regulation and catalytic reaction that explain how the observed allostery works biologically/enzymologically in the system should be provided and fully discussed.

Answer: We thank for this highly relevant fundamental comment. The NanoLuc luciferase (16-point mutant of a catalytic unit of *Oplophorus gracilirostris* luciferase) was engineered as a monomeric enzyme (Hall et al. 2012). In general, all crystallographic structures of NanoLuc luciferase determined so far can be divided into two groups. The first group represents so-called "closed-barrel" NanoLuc structure, where intra-barrel (catalytic cavity) is strongly reduced, preventing luciferin binding to the catalytic site. Instead, there is a ligand-binding pocket shaped on the enzyme surface, where diverse ligand molecules can bind (fatty acid, polyethylene glycols, or luciferins). In our crystallographic experiments, we were able to capture either furimamide or coelenteramide bound in this surface allosteric site. The structures crystallized in the "closed-barrel" NanoLuc form are characterized by tight homotetrameric association, where the bound ligands are found at the protein-protein interface. On the other hand, the second group of NanoLuc structures represents so-called "open-barrel" NanoLuc structure, where the surface allosteric site is disappeared, and instead, a voluminous ligand-binding pocket is formed inside the β -barrel structure. Based on our azacoelenterazine-bound NanoLuc structure, we think that this intra-barrel pocket is a real catalytic site where luciferase reaction occurs. This assumption is now also supported by recently deposited 3-methoxy-furimazine- and inhibitor-bound NanoLuc X-ray structures (PDB IDs: 7SNT and 7SNW; *unpublished data*). In the manuscript revision, we performed additional structure-based mutagenesis in the intra-barrel catalytic site to validate the reaction mechanism that we are proposing in the manuscript. All mutagenesis data are in agreement with structural observation and biochemical conclusions.

Obviously, the NanoLuc luciferase is capable to undergo a large conformational change, between the "closed-barrel" and "open-barrel" structures. We show that a binding to the allosteric site prevents simultaneous binding to the catalytic site, and vice versa. We experimentally evidence that restructuration of the allosteric site can boost the luminescent reaction in the remote intra-barrel active site, demonstrating that the surface allosteric site is vital for enzyme action.

The fact that all crystallographic structures of NanoLuc luciferase in its “closed-barrel” structure show tight homotetrameric association is puzzling. Indeed, it can be a crystallographic artefact, as highlighted by the reviewer. We agree with this concern, and we do not definitively want to over-interpret the observation. What we clearly see is a remarkable conformational transition between the “closed-barrel” and “open-barrel” NanoLuc structure. When adopting the “closed-barrel” conformation, a ligand-binding pocket forms on the enzyme surface. Previous studies showed that this pocket can accommodate a fatty acid molecule. In our experiments, we found bound luciferin molecules, but also PEG molecule, originating from the crystallization buffer. To what extent the crystal packing forces and/or other factors are responsible for the tight homotetrameric association seen in all NanoLuc crystals with the “closed-barrel” is not clear.

During the revision of this manuscript, we performed small-angle X-ray scattering (SAXS) experiments with NanoLuc luciferase in absence or in presence (4 molar excess) of luciferin molecules, in order to probe oligomeric state of NanoLuc luciferase in action. The SAXS experiment showed that the NanoLuc luciferase exists as a monomeric enzyme in standard micromolar concentration, both with and without luciferin molecules. The SAXS results were added into the manuscript. In-solution experiments, in a condition when the NanoLuc is mostly used in diverse bioassays, showed that the enzyme is indeed monomeric. We thus think that the conformational transition (closed-to-open barrel structure) is important for enzyme catalysis and/or its regulation. For instance, the large conformational change can be essential for un/loading of relatively large and hydrophobic luciferin from the intra-barrel catalytic site.

Moreover, we just hypothesize that the conformational transition, coupled with an oligomeric change (monomer-to-tetramer transition captured in several crystal structures), could be a sophisticated mechanism for enzyme catalysis regulation. This could happen *in vivo*, when native *Oplophorus* luciferase is stored in large concentrations in glands at the base of antennae in inactive state. We can imagine that in such “extreme conditions” this could be the case here, to store luciferase/luciferin complexes in inactive states. However, this is only a hypothesis, in order to stimulate future research directions. In our experiments, we worked with engineered NanoLuc luciferase (16 mutations introduced), but not with native *Oplophorus* luciferase that indeed shows complex quaternary structure.

Major findings of this manuscript are: (i) localization of NanoLuc luciferase catalytic site and allosteric site through comprehensive crystallographic experiments, (ii) mapping the molecular recognition between enzyme and luciferin molecules, (iii) delineation of NanoLuc catalytic reaction mechanism, and (iv) highlighting its allosteric behavior, where we yielded improved luciferase variant (NanoLuc^{CTZ}) via an engineering of the allosteric site on the enzyme surface.

As suggested by the reviewer, we provide a hypothetical model for the active-to-inactive switch, to stimulate future directions in the research. The manuscript was modified accordingly, toned down with some conclusions on oligomerization. Our SAXS experiments

showed that the NanoLuc luciferase exists in a solution (at micromolar concentrations) as a monomeric protein.

We added our structure-based hypothesis as a new part to the Results section, in order to synthesize all our results:

A structure-based model for NanoLuc luciferase action

By using X-ray crystallography, we could visualize in this work some luciferin-bound states of NanoLuc luciferase, and thus secure additional insights into the puzzle of its molecular bioluminescence mechanism. From these, along with previously captured crystallographic snapshots as well as available biochemical and computational data, we are providing a structure-based model for this luciferase catalysis (**Fig. 7**). The most striking feature of this model is based on a conformational transition between the so-called “open β -barrel” and “closed β -barrel” structures. Our crystallographic data reveal that the luciferin substrate can bind to an intra-barrel catalytic site (“open β -barrel”) as well as to a secondary allosteric site situated on the surface of the “closed β -barrel” conformation. Moreover, luciferin binding to the allosteric site will prevent its binding to the catalytic site, and *vice versa*, thanks to this conformational transition.

Our model assumes that when the substrate (luciferin) enters the intra-barrel catalytic site (**Fig. 7i**), it is catalytically converted into the product (oxyluciferin) and this is followed by the emission of a blue photon (**Fig. 7ii**). The structure-based mutagenesis experiments described in this work actually confirmed this assumption. Following this, the release of the reaction product from the intra-barrel catalytic site may be facilitated by an open-to-closed conformational transition (**Fig. 7iii**). At this stage, the reaction product may actually shift to the allosteric site appearing on the protein surface upon this conformational change. Our structural and kinetic data indicate that the native oxyluciferin (CEI) uses a radically different binding mode, involving more extensive interactions, with the allosteric site than the synthetic one (FMA). This could explain why our structure-based restructuring of the allosteric site substantially boosted the CTZ-based luminescence, but had no effect on the FMZ-based luminescence. Our kinetic measurements showed that NanoLuc efficiently binds FMZ ($K_m = 0.123 \pm 0.004 \mu\text{M}$) with a minimal product inhibition ($K_p = 0.56 \pm 0.01 \mu\text{M}$), whereas a less effective binding ($K_m = 0.57 \pm 0.02 \mu\text{M}$) and substantial product inhibition ($K_p = 0.256 \pm 0.005 \mu\text{M}$) is observed with native CTZ-luciferin. We suggest that the oxyluciferin product binding to the surface luciferin-binding site provides an allosteric-based negative feedback loop, restraining the β -barrel opening and halting the reaction after several cycles. This would lead to a flash-type bioluminescence, which, unlike bacterial and fungal bioluminescence, is rather typical of marine one. Therefore, we think that the combination of the engineered NanoLuc luciferase with the synthetic FMZ-luciferin reported by Hall and co-workers⁸ actually removed this allosteric-based negative feedback leading to a so-far unmatched glow-type

bioluminescence. Our model also assumes that upon the ligand release from the allosteric site, the free enzyme will regenerate via a closed-to-open β -barrel transition (**Fig. 7iv**).

Finally, NanoLuc luciferase was engineered as an FMZ-using monomeric enzyme⁸, while the corresponding native CTZ-using full-length counterpart, i.e., the catalytic subunit of *Oplophorus* luciferase, displays complex quaternary structure⁷. Here, we have demonstrated with SAXS-based experiments that indeed the NanoLuc exists as a monomeric protein in the micromolar concentration range (**Supplementary Fig. 6**), typical of the conditions of its use in laboratory^{8,12}. Nevertheless, an intriguing feature is that in all the crystallographic structures of NanoLuc, the “closed β -barrel” conformation adopted points out a tightly packed homotetrameric association. This could be a crystallization artifact due to the high concentration of enzyme monomers present in the crystallization drop. However, we hypothesize that this tetramerization may reflect the inherent feature of native *Oplophorus* luciferase although it would have been mostly suppressed in the engineered NanoLuc. In fact, out of the 16 mutations introduced in NanoLuc, up to 7 mutations are localized on the protein surface area involved in the protein-protein contacts seen in the crystallographic tetrameric arrangement. As experimental evidence to support this hypothesis, we constructed and analyzed several NanoLuc “reverse” mutants. As seen by gel filtration, it turned out that some single-point mutants, such as R11E and R43A, did not exist as a pure monomeric protein anymore but rather as monomer-tetramer mixtures (**Supplementary Fig. 10**). We also speculate that in “real life”, such tetrameric complexes could serve as an “inactive” luciferin-loaded storage form of *Oplophorus* luciferase (**Fig. 7v**).

Fig. 7. Structure-based model for NanoLuc luciferase action. I. A substrate molecule enters the intra-barrel catalytic site, the luciferase retains an “open β -barrel” conformation. II. Catalytic conversion of the substrate into its reaction product, followed by the emission of a blue photon. III. Release of the

product out of the intra-barrel catalytic site. This step is accompanied by an open-to-closed conformational transition, and a rebinding of the product to the newly formed allosteric site on the protein surface. IV. Dissociation of the product from the surface allosteric site, allowing recycling into the pre-catalytic “open β -barrel” state. V. Alternatively, the “closed β -barrel” structure of NanoLuc luciferase is permissive to undergo a monomer-to-tetramer transition in crystallization conditions. This step is hypothetical and could only be valid for native *Oplophorus* luciferase which exhibits complex quaternary structure. The PDB ID codes for representative crystallographic structures are provided.

2. Lines 563-566: "This could explain the fact that NanoLuc is monomeric enzyme ... may trigger its homodimerization/homotetramerization ... " Any experimental clues, for example solution scattering or size-exclusion chromatography, were not provided in the presented manuscript. This point largely prevents the findings from being positioned in a bigger picture. This point should be appropriately resolved.

Answer: We thank for this comment, which already relates to the answer in the comment above. In the revision process, we performed additional small-angle X-ray scattering (SAXS) experiments with NanoLuc luciferase in absence and presence (4 molar excess) of luciferin molecule to probe oligomeric state of NanoLuc luciferase in action. The SAXS experiment showed that the NanoLuc luciferase exists as a monomeric enzyme in standard micromolar concentrations, both with and without luciferin molecules. The SAXS results were added into the manuscript. In-solution experiments, in a condition when the NanoLuc is mostly used in diverse bioassays, showed that the enzyme is indeed monomeric. We thus think that the conformational transition is important for enzyme catalysis and/or its regulation. For instance, the large conformational change can be essential for un/loading of relatively large and hydrophobic luciferin from the intra-barrel catalytic site.

We only hypothesize that the conformational transition, coupled with an oligomeric change (monomer-to-tetramer transition captured in several crystal structures), could be a sophisticated mechanism for enzyme catalysis inactivation. This could happen in vivo, when native *Oplophorus* luciferase is stored in large concentrations in glands at the base of antennae in inactive state. We can imagine that in such “extreme conditions” this could be the case here, to store luciferase/luciferin complexes in inactive states. However, this is just a hypothesis. In our experiments, we worked with engineered NanoLuc luciferase (16 mutations introduced), but not with native *Oplophorus* luciferase that indeed shows complex quaternary structure. This is the hypothesis aiming to stimulate future research directions in the field of *Oplophorus* bioluminescence.

3. Lines 688-689: "Concentrated NanoLuc was mixed with a 4 molar excess of furimazine non-oxidizable analogue azaFMZ " This reviewer could not understand the meaning of this procedure. If the azaFMZ molecule was not observed neither in allosteric site nor catalytic center, why this molecule did not bind to the protein? What will happen if this additive was

not provided? Lines 695-696: "The good-looking crystals were soaked overnight in the mother liquor supplemented with 10 mM FMZ or CTZ." It seemed the observed ligands in the crystal structures were added as substrates and reactions took place in the crystallization solutions, and finally the reaction products bound to the allosteric site. If this is correct, this reviewer also did not understand the necessity of this procedure. Why not add the products (CEI or FMA) from the beginning? These points would be very sensitive in interpreting the observed crystal structures, and fully and explicitly explained in the main text.

Answer: We thank for this comment. Here, we clarify our experimental strategy. Indeed, the goal of this crystallographic experiment was to capture azafurimazine (azaFMZ) bound to the enzyme. These efforts yielded diffraction-quality crystals diffracting to high resolutions (up to 1.6 Å). The crystals were built from tightly associated NanoLuc homotetramers, displaying the so-called "closed-barrel" structure. We could find a ligand density in the surface allosteric site, but it was not complete, due to low ligand occupancy. The major issue with azaFMZ is its poor solubility. It is almost impossible to dissolve it completely in co-crystallization experiments. Yes, we mixed protein:ligand in 1:4 molar ratio, but the azaFMZ precipitated heavily, indicating that its active fraction in solution was very low. This observation explains why the occupancy of azaFMZ was low, preventing unambiguous interpretation of its map.

In a next step, to increase the ligand occupancy and yield better-quality map, we decided to soak the crystals in the presence of different luciferin molecules in overnight experiments. However, the soaking with aza-luciferins (azaFMZ and azaCTZ) failed. The crystals did not diffract at all, the soaking with aza-luciferins damaged crystal lattices. On the other hand, the soaking with standard luciferins (FMZ and CTZ), yielded diffraction-quality crystals, and ligand maps could be resolved. There are two reasons why we see oxidized form of luciferins (coelenteramide and furimamide) in the co-crystal structures. The first scenario could be that the enzyme converted the substrate molecules into the product, but we are not in favor of this hypothesis. The enzyme active site is located inside the barrel structure, and we do not think that the crystal lattice could accommodate such large conformational changes, allowing closed-to-open transition and back. We think that the second option was the case. The crystallization and soaking experiments were not performed in anoxic condition. This means that luciferin molecules could be spontaneously oxidized (non-catalytically) by molecular oxygen (O₂) during long-term overnight soaking experiments. We think that the latter explanation is more valid for our observations.

The reason why we did not add oxidized forms (products) from the beginning is simple. The solubility of oxidized luciferins (products) is much worse than their intact luciferin counterparts.

Collectively, crystallographic experiments with imidazopyrazine-based luciferins are rather tricky, and this is mainly for their poor solubility, spontaneous oxidization, and radical intermediates formation. For example, the deciphering of molecular mechanism of the widely used coelenterazine-utilizing *Renilla* luciferase took more than 40 years (Schenk Mayerova et al. 2023, Nature Catalysis). The major obstacle in understanding of its mode of action was a

lack of catalytically “sound” luciferin-bound complexes. In the current manuscript, we performed comprehensive crystallographic experiments again, revealing the nature of luciferin-binding sites in NanoLuc luciferase. We believe that these breakthrough findings will extend our knowledge on NanoLuc luciferase action, and will be critical for engineering of next-generation luciferase-luciferin pairs. Crucially, we demonstrate the importance of our findings by the engineering of the allosteric site, yielding an improved NanoLuc luciferase variant (NanoLuc^{CTZ}) outperforming original NanoLuc luciferase when native, more soluble and less expensive coelenterazine luciferin is used.

Minor:

1. Line 150: "the R1 2-(furan-2-yl) and R3 8-benzyl", line 180: "tail part", line 202: "2-(furan-2-ylmethyl)", or line 203: "8-benzyl", for example. The references to the molecular moieties in this manuscript would be difficult for many of the readers to follow. The molecular formulas for the major ligands, for example CEI, FMA, CTZ, or FMZ, on which mentioned moieties were indicated, should be provided.

Answer: We thank for this comment. We generated and added schemes (**Supplementary Figure 2**).

Supplementary Fig. 2. Schematic representation of bioluminescence reaction catalyzed by luciferase. The chemical structures of the luciferase substrates coelenterazine (a) and furimazine (b) with their oxidized products coelenteramide and furimamide are indicated.

2. Lines 161-162 : "several pocket-shaping residues (e.g. E4, R43 and R166), played a key role during the development of NanoLuc." The pocket-shaping residues should be indicated in Fig. 1i.

Answer: We thank for this comment. The mutated residues during the development of NanoLuc are marked with red dots in Fig. 1i.

3. "Scheme 1" in Suppl Fig. 6 legend was sudden and would get lost in context.

Answer: We thank for this comment. We added a reference to Scheme 1 into the sentence to help the reader to navigate.

4. "CBS-1 and CBS-2" in Fig. 4 legend would be the abbreviation of chloride binding site but not defined anywhere.

Answer: We thank for this comment. We replaced the abbreviations CBS-1 and CBS-2 with the words chloride-binding sites 1 a 2.

5. "contour level 1.0 σ " in Fig. 6a legend. Generally, this threshold is too low. Also the density is not well defined and not convincing that a model based on this density could be used to discuss the reaction coordinates. This figure should be revised with higher threshold ($> 3.0 \sigma$) and the map for entire system including protein must be shown without trimming.

Answer: We thank for this comment. We generated a new version of figure where the map is rendered at 3.0 σ .

6. Fo-Fc maps at catalytic central cavity for the crystal structures of NanoLuc/CEI, NanoLuc/FMA, and NanoLuc-Y94A should be presented to confirm the emptiness of the central cavity.

Answer: We thank for this comment. The map of central catalytic cavity is partially shown in Figure 3c. When the NanoLuc adopts the closed β -barrel, the volume of intra-barrel central cavity is strongly reduced, preventing accommodation of luciferin molecule in catalytically "sound" conformation, as also demonstrated by our molecular dynamics studies.

We added the Fo-Fc maps of the catalytic central cavity for the crystal structures of NanoLuc/CEI, NanoLuc/FMA, and NanoLuc-Y94A, as requested (**Supplementary Fig. 7**).

Supplementary Fig. 7. 2Fo-Fc electron density maps (contour level 3.0σ) of the intra-barrel central cavity in NanoLuc/CEI (PDB ID: 8AQ6), NanoLuc/FMA (PDB ID: 8AQI), and NanoLuc-Y94A (PDB ID: 8AQH). Note that all structures represent the so-called closed β -barrel state with eliminated central cavity preventing the luciferin binding inside.

7. The descriptions of reaction in lines 390-406 and Fig. 6e legend are difficult to follow. The schemes in Fig. 6e should be labeled with serial numbers (e.g., 1, 2, 3[~], or i, ii, iii[~]) and the numbers should be referred to in the texts.

Answer: Thank you for this suggestion, which will certainly improve the manuscript. The figure and text were edited accordingly.

8. The first and second tunnels should be indicated on the structure models in Fig. 7a or 7e.

Answer: We thank for this comment. We produced a new **Fig. 6b** with both tunnels indicated.

9. Molecular formulas in lines 626 and 637 were not clear from where these formulas were referred to. The reference points should be explicitly made in the text.

Answer: We thank for this comment. We moved the part describing the synthesis of azaFMZ to **Supplementary Note 1** in Supporting information to make it clearer.

10. Line 693: "azafurimazine" or lines 745-746 "furimazine and coelenterazine" for example. These molecules would be better presented in the corresponding abbreviations.

Answer: We thank for this comment. We replaced these words with corresponding abbreviations wherever possible. The text was modified accordingly.

Reviewer #3:

The authors investigate the widely used engineered luciferase NanoLuc and describe allosteric relations using new structural insights obtained from co-crystal structures with the reaction products of CTZ and FMZ oxidation, and the unreactive substrate analogue aza-coelenterazine. The reaction products are found to bind at an allosteric binding pocket formed at the interface of two NanoLuc domains. In contrast, aza-coelenterazine is bound in the proposed intramolecular, catalytically active binding site. Based on these and previously derived structures, modeling and site-directed mutagenesis, the authors derive and propose a detailed mechanism for substrate conversion of NanoLuc and its allosteric regulation. In addition, novel NanoLuc variants are obtained with enhanced activity with the classical substrate coelenterazine that rival the activity of NanoLuc with its currently used substrate furimazine. The manuscript is well written and structured and provides key insights for the future improvement of NanoLuc and its applications. While it remains to be established whether the allosteric pocket formed at the dimer-interface observed in the X-ray structure is also formed in solution (see also below), the work is clearly important in a sense that it provides evidence for substrate/product controlled switching between a catalytically-active open and a catalytically-inhibited closed active site. The work is therefore recommended for publication in Nature Communications after minor revisions. Comments are listed below.

Major:

1. A key remaining question is whether the NanoLuc dimer observed in X-ray structures is also formed in solution, in particular at the sub/low nM concentrations that NanoLuc is normally used/studied. The modeling studies reported here suggests that substrate/product could act as a molecular glue to promote dimerization, but no experimental evidence is provided (or cited from the literature). Can the authors comment on how to address this question experimentally?

Answer: We thank for this highly relevant fundamental comment. The NanoLuc luciferase (16-point mutant of a catalytic unit of *Oplophorus gracilirostris* luciferase) was engineered as a monomeric enzyme (Hall et al. 2012). In general, all crystallographic structures of NanoLuc luciferase determined so far can be divided into two groups. The first group represents so-called “closed-barrel” NanoLuc structure, where intra-barrel (catalytic cavity) is strongly reduced, preventing luciferin binding to the catalytic site. Instead, there is a ligand-binding pocket shaped on the enzyme surface, where diverse ligand molecules can bind (fatty acid, polyethylene glycols, or luciferins). In our crystallographic experiments, we were able to capture either furimamide or coelenteramide bound in this surface allosteric site. The structures crystallized in the “closed-barrel” NanoLuc form are characterized by tight homotetrameric association, where the bound ligands are found at the protein-protein interface. On the other hand, the second group of NanoLuc structures represents so-called “open-barrel” NanoLuc structure, where the surface allosteric site is disappeared, and instead, a voluminous ligand-binding pocket is formed inside the β -barrel structure. Based on our

azacoelesterazine-bound NanoLuc structure, we think that this intra-barrel pocket is a real catalytic site where luciferase reaction occurs. This assumption is now also supported by recently deposited 3-methoxy-furimazine- and inhibitor-bound NanoLuc X-ray structures (PDB IDs: 7SNT and 7SNW; *unpublished data*). In the manuscript revision, we performed additional structure-based mutagenesis in the intra-barrel catalytic site to validate the reaction mechanism that we are proposing in the manuscript. All mutagenesis data are in agreement with structural observation and biochemical conclusions.

Obviously, the NanoLuc luciferase is capable to undergo a large conformational change, between the “closed-barrel” and “open-barrel” structures. We show that a binding to the allosteric site prevents simultaneous binding to the catalytic site, and vice versa. We experimentally evidence that restructuring of the allosteric site can boost the luminescent reaction in the remote intra-barrel active site, demonstrating that the surface allosteric site is vital for enzyme action.

The fact that all crystallographic structures of NanoLuc luciferase in its “closed-barrel” structure show tight homotetrameric association is puzzling. Indeed, it can be a crystallographic artefact, as concerned by the reviewer. To dissect this enigmatic feature, during the revision process, we performed small-angle X-ray scattering (SAXS) experiments with NanoLuc luciferase in absence and presence (4 molar excess) of luciferin molecule to probe oligomeric state of NanoLuc luciferase in action. The SAXS experiment showed that the NanoLuc luciferase exists as a monomeric enzyme in standard micromolar concentration, both with and without luciferin molecules. The SAXS results were added into the manuscript. In-solution experiments, in a condition when the NanoLuc is mostly used in diverse bioassays, showed that the enzyme is indeed monomeric. We thus think that the conformational transition is important for enzyme catalysis and/or its regulation. For instance, the large conformational change can be essential for un/loading of relatively large and hydrophobic luciferin from the intra-barrel catalytic site.

Moreover, we hypothesize that the conformational transition, coupled with an oligomeric change (monomer-to-tetramer transition captured in several crystal structures), could be a sophisticated mechanism for enzyme catalysis inactivation. This could happen *in vivo*, when native *Oplophorus* luciferase is stored in large concentrations in glands at the base of antennae in inactive state. We can imagine that in such “extreme conditions” this could be the case here, to store luciferase/luciferin complexes in inactive states. However, this is just a hypothesis. In our experiments, we worked with engineered NanoLuc luciferase (16 mutations introduced), but not with native *Oplophorus* luciferase that indeed shows complex quaternary structure.

Major findings of this manuscript are: (i) localization and atomistic description of NanoLuc luciferase catalytic site (primary luciferin binding site) and allosteric (secondary) site through comprehensive crystallographic experiments, (ii) mapping the molecular recognition between enzyme and luciferin molecules, (iii) delineation of NanoLuc catalytic reaction mechanism, and (iv) highlighting its allosteric behavior, where we yielded improved luciferase variant (NanoLuc^{CTZ}) via an engineering of the allosteric site on the enzyme surface.

Additionally, we provide a hypothetical model for the active-to-inactive switch, to stimulate future directions in the research. The manuscript was modified accordingly, toned down with some conclusions on oligomerization. Our SAXS experiments showed that the NanoLuc luciferase exists in a solution (micromolar concentrations) as a monomeric protein.

We added our structure-based hypothesis as a new part to the Results section, in order to synthesize all our results:

A structure-based model for NanoLuc luciferase action

By using X-ray crystallography, we could visualize in this work some luciferin-bound states of NanoLuc luciferase, and thus secure additional insights into the puzzle of its molecular bioluminescence mechanism. From these, along with previously captured crystallographic snapshots as well as available biochemical and computational data, we are providing a structure-based model for this luciferase catalysis (**Fig. 7**). The most striking feature of this model is based on a conformational transition between the so-called “open β -barrel” and “closed β -barrel” structures. Our crystallographic data reveal that the luciferin substrate can bind to an intra-barrel catalytic site (“open β -barrel”) as well as to a secondary allosteric site situated on the surface of the “closed β -barrel” conformation. Moreover, luciferin binding to the allosteric site will prevent its binding to the catalytic site, and *vice versa*, thanks to this conformational transition.

Our model assumes that when the substrate (luciferin) enters the intra-barrel catalytic site (**Fig. 7i**), it is catalytically converted into the product (oxyluciferin) and this is followed by the emission of a blue photon (**Fig. 7ii**). The structure-based mutagenesis experiments described in this work actually confirmed this assumption. Following this, the release of the reaction product from the intra-barrel catalytic site may be facilitated by an open-to-closed conformational transition (**Fig. 7iii**). At this stage, the reaction product may actually shift to the allosteric site appearing on the protein surface upon this conformational change. Our structural and kinetic data indicate that the native oxyluciferin (CEI) uses a radically different binding mode, involving more extensive interactions, with the allosteric site than the synthetic one (FMA). This could explain why our structure-based restructuring of the allosteric site substantially boosted the CTZ-based luminescence, but had no effect on the FMZ-based luminescence. Our kinetic measurements showed that NanoLuc efficiently binds FMZ ($K_m = 0.123 \pm 0.004 \mu\text{M}$) with a minimal product inhibition ($K_p = 0.56 \pm 0.01 \mu\text{M}$), whereas a less effective binding ($K_m = 0.57 \pm 0.02 \mu\text{M}$) and substantial product inhibition ($K_p = 0.256 \pm 0.005 \mu\text{M}$) is observed with native CTZ-luciferin. We suggest that the oxyluciferin product binding to the surface luciferin-binding site provides an allosteric-based negative feedback loop, restraining the β -barrel opening and halting the reaction after several cycles. This would lead to a flash-type bioluminescence, which, unlike bacterial and fungal bioluminescence, is rather typical of marine one. Therefore, we think that the combination of the engineered NanoLuc luciferase with the synthetic FMZ-luciferin reported by Hall and co-workers⁸ actually removed

this allosteric-based negative feedback leading to a so-far unmatched glow-type bioluminescence. Our model also assumes that upon the ligand release from the allosteric site, the free enzyme will regenerate via a closed-to-open β -barrel transition (**Fig. 7iv**).

Finally, NanoLuc luciferase was engineered as an FMZ-using monomeric enzyme⁸, while the corresponding native CTZ-using full-length counterpart, i.e., the catalytic subunit of *Oplophorus* luciferase, displays complex quaternary structure⁷. Here, we have demonstrated with SAXS-based experiments that indeed the NanoLuc exists as a monomeric protein in the micromolar concentration range (**Supplementary Fig. 6**), typical of the conditions of its use in laboratory^{8,12}. Nevertheless, an intriguing feature is that in all the crystallographic structures of NanoLuc, the “closed β -barrel” conformation adopted points out a tightly packed homotetrameric association. This could be a crystallization artifact due to the high concentration of enzyme monomers present in the crystallization drop. However, we hypothesize that this tetramerization may reflect the inherent feature of native *Oplophorus* luciferase although it would have been mostly suppressed in the engineered NanoLuc. In fact, out of the 16 mutations introduced in NanoLuc, up to 7 mutations are localized on the protein surface area involved in the protein-protein contacts seen in the crystallographic tetrameric arrangement. As experimental evidence to support this hypothesis, we constructed and analyzed several NanoLuc “reverse” mutants. As seen by gel filtration, it turned out that some single-point mutants, such as R11E and R43A, did not exist as a pure monomeric protein anymore but rather as monomer-tetramer mixtures (**Supplementary Fig. 10**). We also speculate that in “real life”, such tetrameric complexes could serve as an “inactive” luciferin-loaded storage form of *Oplophorus* luciferase (**Fig. 7v**).

Fig. 7. Structure-based model for NanoLuc luciferase action. I. A substrate molecule enters the intra-barrel catalytic site, the luciferase retains an “open β -barrel” conformation. II. Catalytic conversion of

the substrate into its reaction product, followed by the emission of a blue photon. III. Release of the product out of the intra-barrel catalytic site. This step is accompanied by an open-to-closed conformational transition, and a rebinding of the product to the newly formed allosteric site on the protein surface. IV. Dissociation of the product from the surface allosteric site, allowing recycling into the pre-catalytic “open β -barrel” state. V. Alternatively, the “closed β -barrel” structure of NanoLuc luciferase is permissive to undergo a monomer-to-tetramer transition in crystallization conditions. This step is hypothetical and could only be valid for native *Oplophorus* luciferase which exhibits complex quaternary structure. The PDB ID codes for representative crystallographic structures are provided.

2. In the absence of experimental verification, the observed binding pockets could still be the results of the very high concentration of NanoLuc monomers in the crystal/ crystallization artefact. The results of the various mutagenesis studies provide more insight into the importance of several key residues and even yield some very useful improved variants, but by themselves do not unequivocally prove the existence of the dimer in solution. The observed conformational changes may still be relevant and hint at a regulatory role in the monomer in solution. It may be good for the authors to discuss the possibility that products are also inhibiting or preferably binding the surface pocket to allosterically regulate the open/close conformational change. Are there any reports in literature that describe product inhibition of NanoLuc?

Answer: We thank for this brilliant comment. Now, we know that the NanoLuc luciferase exists as a monomeric enzyme in micromolar concentrations, in absence and in presence of luciferins (new SAXS data). Single-point mutations in the surface ligand-binding site substantially and selectively boosted the bioluminescence with native CTZ luciferin, but not with the synthetic FMZ luciferin. Indeed, our data point to that the ligand-binding site on the enzyme surface is vital for enzyme catalysis. Mutagenesis experiments also complement crystallographic structures, where native luciferin binds to the surface pocket in a radically different mode than the synthetic one does. We also think that the conformational transition is functionally relevant, with a possible regulatory role. Interesting feature that we reveal is indeed a significant product inhibition seen at NanoLuc-CTZ pair, while the NanoLuc-FMZ pair does not displays this (see Table 1).

3. It would be helpful if the authors could discuss the implications of their findings for biosensor design and other applications of NanoLuc. E.g. the authors write that the extension of NanoLuc C-terminus by two alanines ‘substantially decreased’ CTZ luminescence. Does this imply that C-terminal fusion proteins of this variant, or the new NanoLucCTZ variant, may show reduced luciferase activity when using CTZ? Is this consistent with literature (NanoLuc fusion proteins?) or has this not been described before? What are the implications of their finding for applications involving the split luciferase NanoBiT system?

Answer: Thank you for this comment. Yes, some our mutants, including C-terminus extension or truncation, showed decreased bioluminescence. These mutants displayed approx. 5-fold decrease of the activity, highlighting that the C-terminal fusions can show reduced activity. The activity drop is not dramatic, the proteins still retains high activity when comparing with other luciferase systems. This is very good question, but we are not aware of any comprehensive study that would address how N- or C-terminal fusions can affect luciferase activity. Finally, we believe that with the knowledge gained it will be possible to design a new wave of split-luciferase systems in a more rational manner.

Minor:

1. Line 112/113 renewable light-producing technologies. Why would this help engineering renewable light-producing technologies? How would these look like?

Answer: Thank you for this comment. There are efforts to use bioluminescent organisms or luciferase-based systems as an alternative to traditional sources that rely on electricity, mostly still generated from burning of fossil fuels. For example, implementation of bioluminescent systems, including luciferin synthetic and recycling pathways together with corresponds luciferase genes, into plants and trees it would be possible to construct light-emitting living “biolamps and/or biostreetlights”. For example, such plants or trees could be used for lighting in interiors and exteriors, reducing the need for electricity and providing a unique, eco-friendly and sustainable lighting sources. Such proof of concept results were published in Nature Biotechnology (<https://www.nature.com/articles/s41587-020-0500-9>).

2. Line 171 Figure 1 F: The chord plot is simply mentioned but the main message is not explained in the caption. A short additional explanation in one sentence would be helpful, as it is also very condensed in the main text.

Thank you for this comment. We extended the description of the chord plot in the figure caption to inform that it shows interactions at the the secondary structure level.

3. Figure 3b. No added value in shoing the numbers of kcat and Km that are already presented in Table 1 also as (3 different!) graphs.

Thank you for this comment. In the revised manuscript, **Fig. 3** and **Fig. 5** have been merged into a single **Fig. 4**, which no longer contains the graph in question.

4. Line 242 ‘mechanisms of communication’ sounds a bit broad, potentially a better phrasing would be ‘mechanism of allosteric importance’?

Thank you for this comment. We replaced the term "mechanism of communication" with "mechanism of allosteric interplay".

5. Line 259/260 Why is the flipping of beta strand S5 called 'unusual'? Because it requires an extreme conformational change? Are there any comparable cases that show a beta strand flipping?

It's unusual, because flipping of the beta strand is not common. However, there are a few known cases, for example: (i) In the structure of beta-amyloid peptide, there is a region known as the β -hairpin turn that can undergo a flipping motion. This structural rearrangement is believed to contribute to the aggregation and fibril formation observed in Alzheimer's disease; (ii) In some cases, the hinge region of the IgG molecule, which connects the Fab and Fc regions, can undergo strand flipping. This flipping motion alters the relative positions of the two domains, potentially affecting the antibody's binding properties and interaction with other immune system components.

6. Line 264 change to enzyme-substrate Michaelis complex, as Michaelis complex is a connected term / definition.

Thanks for this comment. We corrected it.

7. Line 281 figure caption (iii) for FMA-bound NanoLuc structure.

Thanks for this comment. We corrected it.

8. Line 324 Might be useful to explain the reason of using EnduRen instead of original coelenterazine? Is EnduRen still cheaper and 'more available' than Endurazine, the used derivate of Furimazine?

Thanks for this comment. EnduRen is a CTZ derivative specifically designed for use in bioluminescent reporter assays. To clarify this, we added this information directly in the text. Luciferin derivatives are standardly used in cell-based assays, as they penetrate better cross cell membranes. The use of intact luciferins in cellular assays is not efficient, because of their low cellular uptake and metabolic instability.

9. For in vitro assays, it would be useful to provide absolute luminescent intensities of both NanoLuc and NanoLucCTZ at the same concentrations and gain, each with both substrates, in vitro to allow a direct comparison of their behaviour. Alternatively, raw data of intensities like photon counts/s should be provided in Fig 5 D, E (line 297)

Thanks for this comment. We refined the data presented in Fig. 5D, E. The values represent number of counts generated by the LuminoCell device. To explain the values to a reader, we added the following lines into the Methods section: “Briefly, luciferase activity was real-time monitored for 24 hours using light-to-frequency converter built in the LuminoCell. Light generated by the luciferase is converted into a series of square-wave pulses, with the frequency depending on the light intensity, thus the luciferase activity is demonstrated by number of detected pulses in a given time (integration time).”

10. Fig 5 D/E: What determines the time-dependence of these signals. It is not clear if and how one compare the values on the Y-axes between d and e, in particular as they are depicted as a.u. However, the scaling between 0 and 100 suggests that some kind of normalization was performed, also because the signal with FMZ is reported to be 50%. The conclusion that the engineered NanoLucCTZ and EnduRen substrate represent a superior luciferase-luciferin reporting pair for long-term live-cell imaging applications require more extensive experiments, e.g, showing similar expression levels for both variants.

Thank you for pointing this out. We do agree that the expression levels of both variants should be determined. Therefore, we assessed the expression of both variants using RT-qPCR approach, see **Supplementary Fig. 18** and updated Methods section. Regarding the data normalization, please see our response above.

11. Unfinished sentence in line 464/465

Thanks for this comment. We corrected it as follows: “The implied timescales plots and Chapman-Kolmogorov tests show two transitions, which hints toward three or more macrostates (**Supplementary Fig. 24**)”.

12. Line 521, write out fatty acid binding protein at first mention of FABP, as done later for nsLTps.

Thanks for this comment. We corrected it.

13. Line 528 mention of initial crystallization experiments. The authors did not provide any data regarding this statement, these could be added to the supporting information.

Thanks for this comment. In this study, we solved 4 NanoLuc structures, first two structures with luciferin bound in the allosteric site, then the Y94A mutant structure, and finally the NanoLucCTZ structure. In the discussion we mention "initial crystallization experiments", however , this is only to point out that we solved the NanoLuc structures with the luciferin

bound in the allosteric site first. We did not perform any other crystallization experiments in this work.

14. Line 564 NanoLuc is a monomeric...

Thanks for this comment. We corrected it.

15. Discussion line 568/569: Interesting hypothesis, but how would the luminescent agents storage be generated in the first place if at lower concentrations the substrate is converted? In additional presence of very high chloride concentration/halide ions? Although this is mentioned later in the paragraph, the connection between these statements is not very clear.

Thanks for this comment. As experimental evidence to support this hypothesis, we constructed and analyzed several NanoLuc “reverse” mutants. As seen by gel filtration, it turned out that some single-point mutants, such as R11E and R43A, did not exist as a pure monomeric proteins anymore but rather as monomer-tetramer mixtures (**Supplementary Fig. 10**). We thus think that native *Oplophorus* luciferase is more prone to self-associate, forming complex quaternary structures (e.g. homotetramers), perhaps presence of chloride anions can serve as additional factor.

Very interesting fact is that the binding of native coelenterazine or coelenteramide, unlike to synthetic furimazine/furimamide, to the allosteric site is radically different. The binding mode of coelenterazine N7 nitrogen in close vicinity to the side chain of K89, which could prevent its initial deprotonation, preventing thus oxidative mechanism. Future experiments with native *Oplophorus* luciferase, but not with engineered NanoLuc that was “monomerized” through several cycles of directed evolution, should answer these questions.

16. Line 579 surface allosteric site could be a ...

Thanks for this comment. We corrected it.

17. Supplement, line 320 reference not listed

Thanks for this comment. The reference list was updated.

18. (Methods) Line 673 Was the expression from 10 ml of 2xLB enough to provide high yields for the NanoLuc variants, or is the large culture not mentioned?

Thanks for this comment. All Nanoluc variants studied in this work had very good expression yields. Typically, we used 100 mL of culture medium to express Nanoluc variants, from which we were able to get more than 10 mg of protein. We corrected the typo of 10 mL to 100 mL.

19. (Methods) Line 684 Please add at least one SDS-PAGE to the supplement to confirm purity of the NanoLuc variants.

Thanks for this comment. The SDS-PAGE gel of different NanoLuc variants was added (**Supplementary Fig. 9**).

20. (Methods) Line 740 Are the mentioned relative light units (RLU) in s⁻¹ M⁻¹ related to the A.U. shown in Fig. 5 (line 297)?

Thanks for this comment. RLUs are the units used in most luminescence measurements that reflect the amount of light produced by the luciferase reaction in vitro as measured by a luminometer. On the other hand, the data in **Fig. 4 f-g** (in revised manuscript) represent measurements of NanoLuc activity in vivo where NanoLuc activity was monitored using the light-to-frequency converter built in the LuminoCell. Light generated by the NanoLuc was converted into a series of square-wave pulses, with the frequency depending on the light intensity, thus the NanoLuc activity is demonstrated by number of detected pulses in a given time (integration time).” Therefore, RLUs and AUs are not related.

REVIEWERS' COMMENTS

Reviewer #1 (Remarks to the Author):

The authors have addressed all my concerns properly and made enough efforts to avoid any overclaiming related to the crystal packing artifact. The added SAXS data also make sense toward what they are claiming. Thus, I believe the manuscript can be published in its current form.

Reviewer #2 (Remarks to the Author):

The authors have sufficiently responded to the comments of this reviewer, and this reviewer recommends this manuscript for publication. One point to be considered in preparing the final manuscript: Figure 7 “V. Inactivation through tetramerization”, expressing tone-down by gray letters is not recommended because just difficult to see. It should be shown in other ways.

Reviewer #3 (Remarks to the Author):

I believe the authors addressed my comments and questions satisfactorily. In particular the presentation and discussion on the role of multimerization improved and the additional SAXS experiments showed that NanoLuc does not form homotetramers in solution. I recommend publication of the manuscript.

Reviewer #4 (Remarks to the Author):

As a reviewer I have been asked specifically to provide a technical assessment of the SAXS component of this work rather than reviewing the manuscript as a whole. The SAXS data would seem to be adequately well measured and fits well the high resolution model of the monomeric enzyme. However, it seems that the SAXS data was provided to address comment number 2 of reviewer 1 who asked for some supporting evidence for a perceived claim of multimerisation of the enzyme. If the SAXS data had found evidence of tetramer this would have addressed the comment but as the SAXS conclusively shows monomer and no change in the presence and absence of luciferin I think the valid comment of reviewer 1 remains unanswered.

In their second comment, reviewer 1 asks for some experimental evidence of multimerisation of the luciferase in response to a perception that the authors claim that multimerisation plays a role in allosteric regulation. In their response to this comment the authors say that: "In the manuscript, we do not claim that the dimerization/tetramerization is crucial for allosteric aspects of the enzyme system." However, as I read the paper I believe that luciferin mediated tetramerisation as a mechanism for regulation is mentioned enough times to make it quite prominent in the findings.

- on page 25: line 619 the authors say : " the presence of the cognate ligand molecule (luciferin) may trigger its self-association in saturated concentration"

- on line 549: "We also speculate that in "real life", such tetrameric complexes could serve as an "inactive" luciferin-loaded storage form".

- Perhaps most prominently the model proposed in figure 7 strongly implies an inactive tetrameric complex.

- on line 630 the authors say: "...NanoLuc inactivation through a closing of its β -barrel structure that is accompanied by an enzyme homooligomerization, as captured in multiple crystal structures.". (Actually, this sentence implies that closing of the beta-barrel is linked to multimerisation and that this is supported by experimental evidence i.e. the crystal packing. Homooligomerisation is not "captured in crystal structures", a crystal is literally one macroscopic homooligomer.)

It is my opinion that the idea of tetramerisation is mentioned enough times in the paper that the authors should provide some direct evidence for it. Someone reading the paper could be forgiven for coming to the conclusion that the paper provides evidence for tetramerisation. I think that this was why reviewer 1 asked for such experimental evidence and I agree. Without experimental evidence for luciferin mediated tetramerisation the authors should remove most of the mentions of tetramerisation perhaps leaving one in the discussion making it clear that it is speculative.

Point-by-point response to comments from all reviewers

Reviewer #1 (Remarks to the Author):

The authors have addressed all my concerns properly and made enough efforts to avoid any overclaiming related to the crystal packing artifact. The added SAXS data also make sense toward what they are claiming. Thus, I believe the manuscript can be published in its current form.

Answer: No comment raised.

Reviewer #2 (Remarks to the Author):

The authors have sufficiently responded to the comments of this reviewer, and this reviewer recommends this manuscript for publication. One point to be considered in preparing the final manuscript: Figure 7 “V. Inactivation through tetramerization”, expressing tone-down by gray letters is not recommended because just difficult to see. It should be shown in other ways.

Answer: We thank for this comment. We accordingly modified the Figure 7.

Reviewer #3 (Remarks to the Author):

I believe the authors addressed my comments and questions satisfactorily. In particular the presentation and discussion on the role of multimerization improved and the additional SAXS experiments showed that NanoLuc does not form homotetramers in solution. I recommend publication of the manuscript.

Answer: No comment raised.

Reviewer #4 (Remarks to the Author):

As a reviewer I have been asked specifically to provide a technical assessment of the SAXS component of this work rather than reviewing the manuscript as a whole. The SAXS data would seem to be adequately well measured and fits well the high resolution model of the monomeric enzyme. However, it seems that the SAXS data was provided to address comment number 2 of reviewer 1 who asked for some supporting evidence for a perceived claim of multimerisation of the enzyme. If the SAXS data had found evidence of tetramer this would have addressed the comment but as the SAXS conclusively shows monomer and no change in the presence and absence of luciferin I think the valid comment of reviewer 1 remains unanswered.

In their second comment, reviewer 1 asks for some experimental evidence of multimerisation of the luciferase in response to a perception that the authors claim that multimerisation plays a role in allosteric regulation. In their response to this comment the authors say that: "In the manuscript,

we do not claim that the dimerization/tetramerization is crucial for allosteric aspects of the enzyme system." However, as I read the paper I believe that luciferin mediated tetramerisation as a mechanism for regulation is mentioned enough times to make it quite prominent in the findings.

- on page 25: line 619 the authors say : " the presence of the cognate ligand molecule (luciferin) may trigger its self-association in saturated concentration"

Answer: We thank for this comment. We agree that the conclusion was too speculative. The sentence was removed from the revised manuscript.

- on line 549: "We also speculate that in "real life", such tetrameric complexes could serve as an "inactive" luciferin-loaded storage form".

Answer: We thank for this comment. The sentence was removed from the revised manuscript.

- Perhaps most prominently the model proposed in figure 7 strongly implies an inactive tetrameric complex.

Answer: We thank for this comment. The Figure 7 schematically depicts proposed structure-based model for NanoLuc luciferase action. The step V. was included to the model to cover all structural observations. The step is hypothetical and could only be valid for native Oplophorus luciferase, which exhibits a complex quaternary structure.

We agree with the reviewer that the step V. is too speculative, and therefore we removed this hypothetical step from the model. Accordingly, the revised manuscript text was modified. We removed all speculative parts.

- on line 630 the authors say: "...NanoLuc inactivation through a closing of its β -barrel structure that is accompanied by an enzyme homooligomerization, as captured in multiple crystal structures.". (Actually, this sentence implies that closing of the beta-barrel is linked to multimerisation and that this is supported by experimental evidence i.e. the crystal packing. Homooligomerisation is not "captured in crystal structures", a crystal is literally one macroscopic homooligomer.)

Answer: We thank for this comment. The sentence was modified to: "Moreover, we identified two chloride-binding sites inside the β -barrel structure, and several additional chloride-binding sites on the enzyme surface, suggesting that chloride ions may contribute to the NanoLuc inactivation through a closing of its β -barrel structure."

It is my opinion that the idea of tetramerisation is mentioned enough times in the paper that the authors should provide some direct evidence for it. Someone reading the paper could be forgiven

for coming to the conclusion that the paper provides evidence for tetramerisation. I think that this was why reviewer 1 asked for such experimental evidence and I agree. Without experimental evidence for luciferin mediated tetramerisation the authors should remove most of the mentions of tetramerisation perhaps leaving one in the discussion making it clear that it is speculative.

***Answer:** We thank the reviewer for his excellent comments. Indeed, we were asked to provide additional experimental data in order to support a hypothesis that the homotetramerization seen in the NanoLuc crystals may reflect behaviour of native Oplophorus luciferase. To address this issue, we constructed and analysed several NanoLuc “reverse” mutants. Importantly, as seen by size-exclusion chromatography experiments, it turned out that some single-point “reverse” mutants, such as R11E and R43A, did not exist as pure monomeric proteins anymore but rather as monomer-tetramer mixtures (**Supplementary Fig. 10**). These mutations are localized on the protein surface area involved in the protein-protein contacts seen in the crystallographic tetrameric arrangement, providing experimental evidence supporting the hypothesis. Therefore, we speculate in the Discussion part that in “real life”, such tetrameric complexes could serve as an “inactive” luciferin-loaded storage form of Oplophorus luciferase.*

*On the other hand, in the revised manuscript we have demonstrated with SAXS-based experiments that indeed the engineered NanoLuc exists as a monomeric protein in the micromolar concentration range (**Supplementary Fig. 6**), typical of the conditions of its use in the laboratory. We did thus our maximum to avoid any misleading or misunderstanding.*

Importantly, our crystallographic studies allowed us to identify and map both the intra-barrel catalytic site and allosteric site shaped on the enzyme surface. Complementary biochemical and biophysical experiments showed that the ligand binding to the surface allosteric site is not the crystallographic artifact, as the restructuration of this site through structure-based mutagenesis confirmed its functional importance. Moreover, an engineered triple mutant, termed as NanoLuc^{CTZ} exhibits superior bioluminescence over the original NanoLuc in cell-based assays, paving new routes for next-generation ultrasensitive bioassays.

To clarify our observations and conclusions, we now modified and/or re-phrased the manuscript parts that were not clear enough and removed all speculative hypotheses from the Results part, as recommended by the reviewer. We agree with the reviewer that it is better to mention this hypothesis only in the Discussion part. Together, we thank for all these valuable comments and suggestions that helped to improve the manuscript.